# Information-Computation Tradeoffs for Noiseless Linear Regression with Oblivious Contamination

**Ilias Diakonikolas**
University of Wisconsin, Madison
ilias@cs.wisc.edu

**Chao Gao**
University of Chicago
chaogao@uchicago.edu

**Daniel M. Kane**
University of California, San Diego
dakane@ucsd.edu

**John Lafferty**
Yale University
john.lafferty@yale.edu

**Ankit Pensia**$^*$
Carnegie Mellon University
ankitp@cmu.edu

## Abstract

We study the task of noiseless linear regression under Gaussian covariates in the presence of additive oblivious contamination. Specifically, we are given i.i.d. samples from a distribution $(x, y)$ on $\mathbb{R}^d \times \mathbb{R}$ with $x \sim \mathcal{N}(0, \mathbf{I}_d)$ and $y = x^\top \beta + z$, where $z$ is drawn independently of $x$ from an unknown distribution $E$. Moreover, $z$ satisfies $\mathbb{P}_E[z = 0] = \alpha > 0$. The goal is to accurately recover the regressor $\beta$ to small $\ell_2$-error. Ignoring computational considerations, this problem is known to be solvable using $O(d/\alpha)$ samples. On the other hand, the best known polynomial-time algorithms require $\Omega(d/\alpha^2)$ samples. Here we provide formal evidence that the quadratic dependence in $1/\alpha$ is inherent for efficient algorithms. Specifically, we show that any efficient Statistical Query algorithm for this task requires VSTAT complexity at least $\tilde{\Omega}(d^{1/2}/\alpha^2)$.

## 1 Introduction

Linear regression is a prototypical supervised learning task with a wide range of applications [RL87; Die01; McD09]. In the vanilla setting, we are given labeled samples $(x^{(i)}, y^{(i)})$, where the covariates $x^{(i)}$ are drawn i.i.d. from a distribution on $\mathbb{R}^d$ and the labels $y^{(i)}$ are (potentially noisy) evaluations of a linear function. The goal of the learner is to approximately recover the hidden regression vector. In this standard setting, linear regression is well-understood both statistically and computationally. Specifically, under Gaussian covariates with additive Gaussian noise, the least-squares estimator is computationally efficient and statistically optimal.

In many real-world scenarios, the input data is subject to some form of contamination, e.g., errors due to skewed and corrupted measurements, making even simple statistical estimation tasks algorithmically challenging. In the context of linear regression, classical computationally efficient estimators inherently fail in the presence of data contamination. An important goal in this context is to understand the possibilities and limitations of computationally efficient estimation in the presence of contaminated data.

---

$^*$The majority of this work was done while the author was at the Simons Institute, UC Berkeley.

39th Conference on Neural Information Processing Systems (NeurIPS 2025).

In this work, we study the fundamental problem of linear regression with Gaussian covariates in the presence of *oblivious* additive contamination in the responses (see Definition 1.1). In the oblivious contamination model, an adversary is allowed to corrupt a $(1-\alpha)$-fraction of the labels (by adding an adversarially selected value to the label), for some parameter $\alpha > 0$, and is limited in their capability by requiring the contamination be *independent* of the samples. Interestingly, the oblivious model information-theoretically allows for consistent estimation even for $\alpha \to 0$. This stands in contrast to the more challenging model of adversarial contamination [Hub64; DK23], where non-trivial guarantees are impossible if more than half of the labels are corrupted.

To facilitate the subsequent discussion, we define our learning task below.

**Definition 1.1** (Noiseless Linear Regression with Oblivious Contamination in Responses). Let $\alpha \in (0, 1)$ be the probability of inliers. Let $E$ be a univariate distribution with $\mathbb{P}_{Z \sim E}(Z = 0) \geq \alpha$. For $\beta \in \mathbb{R}^d$, we denote by $P_{\beta, E}$ the distribution on labeled examples $(x, y) \in \mathbb{R}^d \times \mathbb{R}$ defined as follows:

$$x \sim \mathcal{N}(0, \mathbf{I}_d) \text{ and } y = x^\top \beta + Z, \text{ where } Z \sim E \text{ is independent of } x.$$

Given i.i.d. samples $\{(x_i, y_i)\}_{i=1}^n$ from an unknown $P_{\beta^*, E}$, the goal is to construct an estimate $\widehat{\beta}$ such that $\|\widehat{\beta} - \beta^*\|_2$ is small.

The model of Definition 1.1 goes back to the work of Candes and Tao [CT05], who studied it (for more general design matrices) as a classical example of error correction. It is also a standard model in face recognition [WM10], image inpainting [NT13], privacy-preserving data analysis [DMT07], and model repair [GL20]. A basic result in this area is that the true $\beta$ can be recovered exactly, as long as the design matrix satisfies restricted isometry (therefore, for Gaussian design) and the number of nonzero entries of the noise is not too large (detailed below) [CT05; CRTV05; WM10; NT13; GL20]. Interestingly, Candes and Tao [CT05] noted that the model can also be recast as compressed sensing.

The statistical task of linear regression with Gaussian covariates under oblivious contamination has been extensively studied over the past decade [TJSO14; JTK14; BJK15; BJKK17; SBRJ19; PF20; DT19; dNS21]. The oblivious model has also been explored for other natural tasks, including PCA, sparse recovery [PF20; dLNNST21], and estimating a signal with additive oblivious contamination [dNNS22]. While most prior work has focused on Gaussian or subgaussian design matrices, a more recent line of investigation has developed efficient estimators in the distribution-free setting under mild assumptions [DKPT23a; DKPT23b].

Let us return to Definition 1.1 and discuss the precise quantitative aspects. Ignoring computational constraints, the sample complexity $n$ required to obtain any non-trivial estimate of $\beta^*$ for the problem of Definition 1.1 is $n = d/\alpha$; in fact, $n = \Theta(d/\alpha)$ samples suffice to estimate $\beta^*$ *exactly*. In contrast, the best known computationally efficient algorithms require sample complexity of $n = \Omega(d/\alpha^2)$ samples [GL20; dNS21]. Interestingly, known polynomial-time algorithms using $n = O(d/\alpha^2)$ samples succeed even for the (more challenging) *noisy* version of the estimation task—where (in addition to oblivious contamination) the clean labels are perturbed by random observation noise (e.g., Gaussian noise).[2]

While *noisy* linear regression with oblivious contamination information-theoretically requires $\Omega(d/\alpha^2)$ samples, this is not the case for the *noiseless* version considered in this work—where, as mentioned above, $O(d/\alpha)$ samples suffice. This quadratic gap in $1/\alpha$ between the information-theoretic optimum and the sample complexity of known polynomial-time algorithms can be significant in applications where the fraction of inliers $\alpha$ is small. Beyond practical considerations, given the fundamental nature of this estimation problem, it is natural to ask whether a computationally efficient algorithm with (near-)optimal sample complexity (i.e., within logarithmic factors of the optimal) exists. This leads to the central question motivating our work:

**Question 1.2.** Does there exist a constant $c > 0$ so that for all $d \in \mathbb{N}, \alpha \in (0, 1)$, there exists an algorithm, using $O(\frac{\text{poly}(d)}{\alpha^{2-c}})$ samples and running in $\text{poly}(d, n)$ time, that computes an estimate $\widehat{\beta}$ such that $\|\widehat{\beta} - \beta^*\|_2$ is small?

---

[2]To be precise, in the *noisy* version of the problem, the labels are of the form $y = x^\top \beta + \xi + Z$, where $\xi \sim N(0, \sigma^2)$. Definition 1.1 corresponds to the important special case of $\sigma = 0$. For the noisy case of $\sigma > 0$, the information-theoretic error rate is $\|\widehat{\beta} - \beta^*\|_2 = \Theta(\sigma \cdot \sqrt{\frac{d}{n\alpha^2}})$.

Our main result answers this question in the negative for efficient Statistical Query (SQ) algorithms—a broad and well-studied family of algorithms.

## 1.1 Main Result

To establish our negative result, we shall show that even the following (easier) testing task is computationally hard for SQ algorithms:

**Testing Problem 1.3** (Testing Version of Linear Regression with Oblivious Contamination). Let $\rho > 0$ be the signal strength and $\alpha \in (0, 1)$ be the inlier probability. Let $E$ be a (known) univariate distribution on $\mathbb{R}$ that assigns at least $\alpha$ probability to $0$. Let $R^*_{\rho,E}$ be the univariate distribution of $G + z$, where $G \sim \mathcal{N}(0, \rho^2)$ and $z \sim E$ independently. The algorithm gets sample access to a distribution $(x, y) \sim \Theta$ with the goal of distinguishing:

- "Null": $\Theta = P$, where under $P$: $x \sim \mathcal{N}(0, \mathbf{I}_d)$ and $y \sim R^*_{\rho,E}$ independently.

- "Alternate": First a unit vector $v$ is sampled uniformly, and then conditioned on $v$, $\Theta = Q_v$, where under $Q_v$: $x \sim \mathcal{N}(0, \mathbf{I}_d)$ and $y = \rho v^\top x + z$, where $z \sim E$ is independent of $x$.

We say that an algorithm $\mathcal{A}$ succeeds if the failure probability of $\mathcal{A}$ is less than $1/10$ under both the "null" and the "alternate".

Note that, under the null hypothesis, the features $x$ and the responses $y$ are independent of each other; while under the alternate hypothesis, they follow the distribution $P_{\beta,E}$ of Definition 1.1 with $\|\beta\|_2 = \rho$. We show in Appendix C that a (computationally-efficient) estimation algorithm for the task of estimating $\beta$ with error $\rho/4$ suffices to (computationally-efficiently) solve the testing problem above.

**Proposition 1.4** (Efficient Reduction of Testing to Estimation; Informal). *If there exists a computationally-efficient algorithm to compute $\widehat{\beta}$ with $\|\widehat{\beta} - \beta^*\| \leq \rho/4$ with high probability, then it can be transformed into a computationally-efficient algorithm for Testing Problem 1.3.*

**Basics on SQ Algorithms.** Instead of getting sample access, SQ algorithms [Kea98; FGRVX17] interact with the underlying distribution $D$ through the following oracle.

**Definition 1.5** (VSTAT Oracle). Let $D$ be a distribution on $\mathcal{X}$. A statistical query is a bounded function $f : \mathcal{X} \to [0, 1]$. For a "simulation complexity" $m \in \mathbb{N}$, a VSTAT($m$) oracle for the distribution $D$ on the input $f$ returns a value $v$ such that $|v - \mathbb{E}_D[f]| \leq \max\{1/m, \sqrt{(\mathbb{E}_D[f](1 - \mathbb{E}_D[f]))/m}\}$.

That is, the VSTAT($m$) oracle returns an estimate of $\mathbb{E}_D[f]$ with error comparable to the deviation in Bernstein's inequality for high-probability estimates of taking $m$ i.i.d. samples from the Bernoulli distribution with bias $\mathbb{E}_D[f]$. We thus refer to $m$ as the simulation complexity.

A *Statistical Query (SQ) algorithm* is an algorithm whose objective is to learn some information about an unknown distribution $D$ by making adaptive calls to the corresponding VSTAT oracle. The complexity of an SQ algorithm is quantified by the total number of queries to the VSTAT oracle (viewed as a measure of the algorithm's running time) and the maximum simulation complexity of any such query (viewed as a measure of the algorithm's sample complexity).

In the context of our learning problem (Definition 1.1), it is worth pointing out the following. First, there exists an inefficient SQ algorithm with small simulation complexity, which in particular can be simulated using $\widetilde{O}(d/\alpha)$ many i.i.d. samples (see Appendix D). Second, there exist efficient SQ algorithm whose simulation complexity matches the sample complexity $\widetilde{O}(d/\alpha^2)$ of known efficient algorithms (see Appendix E).

With this context, our main result is the following:

**Theorem 1.6** (SQ Hardness of Testing Problem 1.3; informal). *Consider the Testing Problem 1.3. Suppose that (i) $\alpha \gg \frac{1}{d^{\text{polylog}(d)}}$ (i.e., the fraction of inliers is not too tiny) and (ii) $\rho = \widetilde{\Theta}(\alpha)$. Then there exists a distribution $E$ satisfying $\mathbb{P}_{Z \sim E}(Z = 0) \geq \alpha$ such that any SQ algorithm that solves Testing Problem 1.3 either*

- *uses $d^{\Omega(\log^2(d/\alpha))}$ many queries, or*
- *uses at least one query to VSTAT($m$) for $m = \widetilde{\Omega}(\sqrt{d}/\alpha^2)$.*

Informally speaking, Theorem 1.6 shows that no SQ algorithm can solve the testing problem (and, via Proposition 1.4, the estimation problem of approximating $\beta^*$) with less than super-polynomial in $d$ many queries, unless using queries whose simulation complexity is at least $\widetilde{\Omega}(\sqrt{d}/\alpha^2)$. That is, either the algorithm "uses" $\widetilde{\Omega}(\sqrt{d}/\alpha^2)$ many "samples" (in the sense of simulation complexity mentioned above) or it takes super-polynomial "time" (in the sense of number of queries). We thus obtain evidence that the quadratic dependence in $1/\alpha$ on the sample size is required for computationally efficient algorithms.

It is worth noting that the SQ-hard instances that we construct for the testing problem *are* efficiently solvable with $\widetilde{O}(\sqrt{d}/\alpha^2)$ samples. We conjecture that the correct dependence on $d$ is in fact linear (i.e., an $\Omega(d/\alpha^2)$ lower bound on the computational sample complexity). This is left as an interesting question for future work (see Section 4).

Finally, while the focus of this work is on the SQ model, SQ-hardness results typically translate to quantitatively similar hardness for low-degree polynomial tests [Hop18; KWB19], via the work of [BBHLS21]. While we do not establish a formal theorem in this regard, we believe that our SQ-hard instances are also hard for low-degree polynomials.

## 1.2 Overview of Techniques

We wish to show that it is hard to solve Testing Problem 1.3 with fewer than $\sqrt{d}/\rho^2$ samples (we will ultimately set $\rho = \widetilde{\Theta}(\alpha)$). The first question we face is to make a judicious choice of the contamination distribution $E$ that (I) satisfies our noise model, namely $\mathbb{P}_{Z\sim E}(Z = 0) \geq \alpha$; and (II) it is SQ-hard to distinguish the null and alternate hypotheses.

**Choice of Contamination Distribution: Intuition.** A natural first step to consider is what happens if we select the contamination distribution $E$ to be the standard Gaussian, i.e., $E = \mathcal{N}(0, 1)$. In this case, the testing task corresponding to Testing Problem 1.3 is *information-theoretically* impossible with $o(\sqrt{d}/\rho^2)$ samples. Unfortunately, this choice does not fit our criterion (I), requiring that the contamination distribution must be *exactly* 0 with probability at least $\alpha$.

Inspired from the information-theoretic sample complexity lower bound for the Gaussian contamination setting, we instead consider a scenario where the contamination is given by a distribution $E$, which is a *discrete* Gaussian with spacing $s$ (see Definition 2.9). Heuristically, the discrete Gaussian approximately matches its low-degree moments with the continuous Gaussian case, and thus, prior work [DKS17] hints that it is SQ-hard to distinguish between the cases of discrete Gaussian and continuous Gaussian contamination. Since the case of continuous Gaussian contamination information-theoretically requires $\Omega(\sqrt{d}/\rho^2)$ samples, intuitively we are moving in the right direction. Note that the aforementioned discrete Gaussian $E$ assigns probability $\Omega(s)$ to 0. Taking $s = \Theta(\alpha)$, we simultaneously satisfy criterion (I) above and have a reasonable chance of satisfying (II).

The above is the key intuitive idea underlying our proof. However, there are a number of important technical steps required to make the analysis work towards satisfying (II).

**Discrete Noise and Non-Gaussian Component Analysis.** For a unit vector $v$, let $Q_v$ be the distribution over $(x, y)$ such that $y = \rho v^\top x + Z$, where $Z \sim E$ independently of $x$ and $E$ is the suitable discrete Gaussian distribution. Let $P$ be the distribution over $(x, y)$ corresponding to the null hypothesis, namely $x$ and $y$ are independent with correct marginals (i.e., $x \sim \mathcal{N}(0, \mathbf{I}_d)$ and $y \sim \rho^2 G + Z$, where $G \sim \mathbb{N}(0, 1)$ and $Z \sim E$ are independent). We wish to show that it is SQ-hard to distinguish between $Q_v$, for random $v$, and $P$. Note that conditioning on the value of $y$, $Q_v$ is a standard Gaussian in the directions orthogonal to $v$ and is given by some known distribution, $A_y$, in the $v$–direction. This means that the testing problem we are considering is effectively a *conditional* Non-Gaussian Component Analysis (NGCA) problem (Testing Problem 2.8). Unfortunately, there are several technical obstacles preventing us from applying existing tools [DKS17; DKS19].

The first technical hurdle arises from the fact that $A_y$ is a discrete distribution, and in particular has infinite chi-squared norm with respect to the standard Gaussian. In particular, this means that the standard SQ–dimension related techniques for proving lower bounds will not work here. Instead, we need to leverage and adapt the recent work of [DKRS23] that directly uses Gaussian Fourier analysis to establish SQ lower bounds even when the chi-squared distance is infinite. Unfortunately, the latter

work [DKRS23] does not give SQ-lower bounds for *conditional* Non-Gaussian Component Analysis tasks (as the one we are dealing with here). Consequently, we will require a careful adaptation of their techniques in our context.

**Connection with Continuous Gaussian Contamination.** A key requirement for the Gaussian Fourier analysis to go through in [DKRS23] is that $A_y$'s have well-behaved moments. Unfortunately, an additional technical challenge arising in our context is that directly bounding the relevant moments of $A_y$ (which belongs to the family of discrete Gaussians) is challenging.

Instead, for the purpose of the analysis, we again leverage the connection with continuous Gaussian contamination. Specifically, we choose $B_y$ to be a continuous Gaussian counterpart of the discrete Gaussian $A_y$. Let the resulting distribution on $(x, y)$ be $T_v$ (which is a continuous counterpart of $Q_v$). Note that this is again an instance of conditional NGCA. Since the $B_y$'s are now (continuous) Gaussians (and hence satisfy many desirable properties, e.g., continuity), it can be shown that if $v$ and $w$ are nearly orthogonal vectors, $T_v$ and $T_w$ will have small chi-squared inner product with respect to $P$.

**Hardness of Continuous Noise Contamination.** The fact that two random unit vectors have small inner product with high probability can be used to show that the task of testing between $P$ and $\{T_v\}_{v \sim \mathcal{S}^{d-1}}$ has large SQ dimension. This implies SQ-hardness of this basic testing problem. In fact, it will imply the more powerful result that for any bounded function $f$, with high probability over $v$, the expectations $\mathbb{E}_{T_v}[f]$ and $\mathbb{E}_P[f]$ cannot be distinguished by a VSTAT$(o(m_0))$ query for $m_0 := \sqrt{d}/(\rho^2 \cdot \log^4 d)$; see Proposition 3.6.

**Quantitative Relationship between Discrete and Continuous Gaussian Noise.** We now return to the challenge of computing moments of discrete Gaussians $A_y$ (for performing Gaussian Fourier analysis). We resolve this issue by comparing these moments to the moments of $B_y$. As $A_y$ will be a discrete version of the Gaussian $B_y$, this relationship will be relatively manageable to prove. We then combine this ingredient with techniques involving Hermite analysis from [DKRS23] to show the following: for any bounded test function $f$, with high probability over the choice of a random $v$, it holds that $|\mathbb{E}_{Q_v}[f] - \mathbb{E}_{T_v}[f]|$ is tiny (inverse super-polynomial in $m_0$) as long as $s \ll \frac{\rho}{\text{polylog}(d)}$ (Theorem 3.7).

**Putting Everything Together.** Combining the above, we obtain the following: for any $f$, with high probability over random $v$, it holds that (i) $|\mathbb{E}_{Q_v}[f] - \mathbb{E}_{T_v}[f]|$ is inverse super-polynomially small in $m_0$, and (ii) $|\mathbb{E}_{T_v}[f] - \mathbb{E}_P[f]|$ is smaller than the threshold for VSTAT$(o(m_0))$. Therefore, by a union bound and a triangle inequality, it follows that with high probability $|\mathbb{E}_{Q_v}[f] - \mathbb{E}_P[f]|$ is also smaller than the threshold for VSTAT$(o(m_0))$, implying SQ-hardness (Proposition 2.6).

## 1.3 Related Work

Our work is broadly situated in the field of robust statistics, which has a long history dating back to Huber and Tukey [Hub64; Tuk60]. Robust statistics aims to design estimators that are tolerant to data contamination. Focusing on high-dimensional data, our work studies the statistical and computational aspects of robust estimation, which has seen a flurry of work in the last decade since [DKKLMS16; LRV16]; see [DK23] for a recent book on this topic. For designing robust estimators, the choice of contamination model naturally plays a crucial role. This work is part of a broader effort to understand computational and statistical aspects of natural, not fully adversarial, contamination models; see, e.g., [BJK15; BJKK17; ZJS19; DGT19; DK22; DKMR22; DKRS22; DKKTZ22; DKPT23a; DDKWZ23b; DDKWZ23a; MVBWS24; NGS24; PP24; DZ24; KG25; DIKP25].

Historically, the prototypical contamination model in robust statistics has been Huber's contamination model [Hub64], which was strengthened to total variation distance [Hub65] and strong contamination models [DKKLMS16]. The task of linear regression under these contamination models is now well understood both statistically [CGR16] and computationally [DKS19; PJL20; DKPP23]. As mentioned earlier, it is information-theoretically impossible to achieve consistency in these models if the proportion of contamination is bounded away from zero. Thus, an important direction is to understand the possibilities and limitations in other, less adversarial, contamination models. The *oblivious* adversary studied here is one such model, and indeed it does lead to consistent estimation even when the oblivious outliers constitute the majority of the observed data; see the discussion below Defini-

 Our work shows that while this weaker contamination model is benign from the perspective of information-theoretic rates, it does present surprising information-computation tradeoffs.

## 2  Preliminaries

For a univariate distribution $E$, we define $R^*_{\rho, E}$ to be the univariate distribution of $G + z$, where $G \sim \mathcal{N}(0, \rho^2)$ and $z \sim E$ independently. For two vectors $v$ and $w$ in $\mathbb{R}^d$, we use $\langle v, w \rangle$ to denote the standard inner product $\sum_{i \in [d]} v_i w_i$. A degree-$k$ tensor in $d$-dimensions $\mathbf{v}$ is an element in $(\mathbb{R}^d)^{\otimes k}$. with entries $(\mathbf{v}_{i_1, \ldots, i_k})_{i_1 \in [d], \ldots, i_k \in [d]}$. For a vector $v$, we use $v^{\otimes k}$ to denote the $k$-tensor with entries $\prod_{\ell = 1}^d v_{i_\ell}$. For two $k$-tensors $\mathbf{v}$ and $\mathbf{w}$, we use $\langle \mathbf{v}, \mathbf{w} \rangle$ to denote the inner product $\sum_{i_1, \ldots, i_k} \mathbf{v}_{i_1, \ldots, i_k} \mathbf{w}_{i_1, \ldots, i_k}$ and use $\|\mathbf{v}\|_2 := \sqrt{\langle \mathbf{v}, \mathbf{v} \rangle}$. A $k$-tensor function $\mathbf{F} : \mathcal{X} \to (\mathbb{R}^d)^{\otimes k}$ maps each $x \in \mathcal{X}$ to a $k$-tensor.

**Hermite Polynomials** For a $k \in \mathbb{N}$, we use $h_k : \mathbb{R} \to \mathbb{R}$ to denote the $k$-th normalized probabilist's polynomial (which is a degree-$k$ polynomial with definition $h_k(x) := \frac{1}{\sqrt{k!}}(-1)^k e^{x^2/2} \frac{d^k}{dx^k} e^{-x^2/2}$). We shall also use the $k$-th Hermite tensor $\mathbf{H}_k$ as defined in [DKRS23, Definition 2.2].

**Fourier Analysis** For a distribution $P$ on a domain $\mathcal{X}$, we use $L^2(\mathcal{X}, P)$ to denote the space of all functions $f : \mathcal{X} \to \mathbb{R}$ with $\mathbb{E}_{x \sim P}[f^2(x)] < \infty$. For two functions $f, g \in L^2(\mathcal{X}, P)$, we use $\langle f, g \rangle_P$ to denote the inner product $\mathbb{E}_{x \sim P}[f(x)g(x)]$ and $\|f\|_{L_2(P)}$ to denote $\langle f, f \rangle_P$. For a function $f : \mathbb{R}^d \to \mathbb{R}$ and an $\ell \in \mathbb{N}$, we define $f^{\leq \ell}$ to be the degree-$\ell$ Hermite approximation function $f^{\leq \ell}(x) := \sum_{k=0}^\ell \langle \mathbf{A}_k, \mathbf{H}_k \rangle$ where $\mathbf{A}_k := \mathbb{E}_{x \sim P}[f(x)\mathbf{H}_k(x)]$. We extend this definition to $f : \mathbb{R}^d \times \mathbb{R}$ as follows: First, for each $y \in \mathbb{R}$, we define $f_y : \mathbb{R}^d \to \mathbb{R}$ as $x \mapsto f(x, y)$ and then define $f^{\leq \ell}(x, y) := f_y(x)^{\leq \ell}$, that is, for each $y$, we perform degree-$\ell$ approximation of $f_y$. We use $f^{> \ell} := f - f^{\leq \ell}$ to denote the residual.

**Fact 2.1.** For every function $f : \mathbb{R}^d \to [-1, 1]$, $\|f^{> \ell}\|_{L_2(\mathcal{N}(0, \mathbf{I}_d))} \to 0$ as $\ell \to \infty$. Furthermore, for all $f : \mathbb{R}^d \times \mathbb{R} \to [-1, 1]$ and univariate measures $R$, $\|f^{> \ell}\|_{L_2(\mathcal{N}(0, \mathbf{I}_d) \times R)} \to 0$ as $\ell \to \infty$.

We use $\widetilde{\Omega}, \widetilde{\Theta}$ notation to hide $\mathrm{polylog}(d, 1/\alpha)$ factors. For two non-negative functions, $a$ and $b$, we use $a \lesssim b$ (similarly $a \gtrsim b$) to say that there exists a constant (independent of other problem parameters) such that $a \leq Cb$ (respectively, $a \geq Cb$); if $a \lesssim b$ and $b \gtrsim a$, then we say $a \asymp b$.

**SQ Algorithms** We state the preliminaries of SQ for the following generic testing problem.

**Testing Problem 2.2** (Generic Testing Problem). Let $\mathcal{P}$ and $\{\mathcal{Q}_v\}_{v \in \mathcal{S}^{d-1}}$ be distributions over a domain $\mathcal{Z}$, which correspond to "null" and "alternate", respectively.

- First sample $\Gamma \sim \mathrm{Ber}(1/2)$ and $v \sim \mathcal{S}^{d-1}$ independently (unknown to the statistician).
- Then set $\Theta = \mathcal{P}$ ("null") if $\Gamma = 0$ and $\Theta = \mathcal{Q}_v$ ("alternate") otherwise.
- The statistician gets (either sample/oracle) access to the distribution $\Theta$ and generates $\widehat{\Gamma} \in \{0, 1\}$ using an algorithm $\mathcal{A}$. We say $\mathcal{A}$ solves the testing problem if $\mathbb{P}(\widehat{\Gamma} \neq \Gamma) \leq 0.1$.

We say an SQ algorithm $\mathcal{A}$ solves a problem with query complexity $q$ and accuracy complexity $m$ if it iteratively (potentially also adaptively and randomly) makes queries $f_1, \ldots, f_q$ (each $f_i$ is bounded in $[0, 1]$ and could depend on the previous queries and their responses) on the underlying distribution ($\Theta$ above) to a $\mathrm{VSTAT}(m)$ oracle. An SQ lower bound is an information-theoretic lower bound of the following form: any successful SQ algorithm $\mathcal{A}$ must have either $q \geq q_0$ or $m \geq m_0$. In the remainder of this section, we detail the technical results for proving such lower bounds.

**Definition 2.3** (Pairwise Correlation). For a reference distribution $\mathcal{P}$, and candidate distributions $\mathcal{Q}_1$ and $\mathcal{Q}_2$, the pairwise correlation between $\mathcal{Q}_1$ and $\mathcal{Q}_2$ with respect to $\mathcal{P}$ is defined as $\chi_{\mathcal{P}}(\mathcal{Q}_1, \mathcal{Q}_2) := \mathbb{E}_{Z \sim \mathcal{P}}\left[\frac{q_1(Z)q_2(Z)}{p^2(Z)} - 1\right]$, where $q_1(\cdot), q_2(\cdot), p(\cdot)$ denote the densities of $\mathcal{Q}_1, \mathcal{Q}_2$, and $\mathcal{P}$ with respect to a common measure, respectively. When $\mathcal{Q}_1 = \mathcal{Q}_2$, the pairwise correlation becomes the same as the $\chi^2$-divergence between $\mathcal{Q}_1$ and $\mathcal{P}$, i.e., $\chi^2(\mathcal{Q}_1, \mathcal{P}) = \int_{\mathcal{Z}} \frac{q_1^2(x)}{p(x)} dx - 1$.

Statistical dimension is then defined using these pairwise correlations:

**Definition 2.4** (Statistical dimension from [BBHLS21]). The statistical dimension of Testing Problem 2.2 at accuracy complexity $m$ is defined as:

$$\mathrm{SDA}(m) := \max\left\{q \in \mathbb{N} : \sup_{\mathcal{E}:\mathbb{P}_{v,v'}(\mathcal{E}) \geq 1/q^2} \mathbb{E}_{v,v'}\left[\left|\chi_{\mathcal{P}}(\mathcal{Q}_v, \mathcal{Q}_{v'})\right| \big| \mathcal{E}\right] \leq \tfrac{1}{m}\right\},$$

where (i) $v, v' \overset{\mathrm{iid}}{\sim} \mathcal{S}^{d-1}$ independently and (ii) the inner supremum is taken over events $\mathcal{E} \subset \mathcal{S}^{d-1} \times \mathcal{S}^{d-1}$ on $v$ and $v'$ (i.i.d. from unit sphere) that have probability at least $1/q^2$.

We now define the notion of success of a query $f$ that will be useful to us:

**Definition 2.5** (Success of a query on a distribution)**.** We say that a query $f : \mathcal{Z} \to [0, 1]$ succeeds on distinguishing $\mathcal{Q}_v$ and $\mathcal{P}$ with accuracy complexity $m$, denoted by the event $\mathcal{E}_{f,v,m}$, if $|\mathbb{E}_{\mathcal{Q}_v}[f(Z)] - \mathbb{E}_{\mathcal{P}}[f(Z)]| \geq \max\left(\frac{1}{m}, \min\left(\sqrt{\frac{a(1-a)}{m}}, \sqrt{\frac{b(1-b)}{m}}\right)\right)$ for $a := \mathbb{E}_{\mathcal{P}}[f]$ and $b := \mathbb{E}_{\mathcal{Q}_v}[f]$.

We are now equipped to state the SQ lower bounds that we will use repeatedly.

**Proposition 2.6** (SQ Lower Bound)**.** *Consider Testing Problem 2.2. Then*

- *(C.I) For any query $f : \mathcal{Z} \to [0, 1]$, $\mathbb{P}_{v \sim \mathcal{S}^{d-1}}\left(\mathcal{E}_{f,v.m}\right) \leq \frac{1}{\mathrm{SDA}(7m)}$.*

- *(C.II) Suppose for all queries $f : \mathcal{Z} \to [0, 1]$, it holds that $\mathbb{P}_{v \sim \mathcal{S}^{d-1}}\left(\mathcal{E}_{f,v,m}\right) \leq \frac{1}{q}$. Then any SQ algorithm $\mathcal{A}$ that solves Testing Problem 2.2 must use either $\Omega(q)$ queries in expectation or at least one query as powerful as VSTAT$(m+1)$.*

**Non-Gaussian Component Analysis** We will primarily consider Testing Problem 2.2 of a particular form called Non-Gaussian Component Analysis (NGCA). We begin by defining High-Dimensional Hidden Direction Distribution:

**Definition 2.7** (High-Dimensional Hidden Direction Distribution)**.** For a unit vector $v \in \mathbb{R}^d$ and a distribution $\mathcal{H}$ on the real line, we define $P_v^{\mathcal{H}}$ to be the distribution over $\mathbb{R}^d$, where $P_v^{\mathcal{H}}$ is the product distribution whose orthogonal projection onto the direction of $v$ is $\mathcal{H}$, and onto the subspace perpendicular to $v$ is the standard $(d-1)$-dimensional normal distribution. In particular, if $\mathcal{H}$ is a continuous distribution with probability density function (pdf) $\mathcal{H}(x)$, then $P_v^{\mathcal{H}}(x)$ has the pdf $\mathcal{H}(v^\top x)\phi_{\perp v}(x)$, where $\phi_{\perp v}(x) = \exp\left(-\|x - (v^\top x)v\|_2^2/2\right)/(2\pi)^{(d-1)/2}$.

[DKS17] established SQ lower bounds for the NGCA problem, where the null and the alternate are $\mathcal{N}(0, \mathbf{I}_d)$ and $\{P_v^{\mathcal{H}}\}_{v \sim \mathcal{S}^{d-1}}$, respectively and (i) $\mathcal{H}$ (nearly) matches many moments with $\mathcal{N}(0, 1)$ and (ii) has finite $\chi^2(\mathcal{H}, \mathcal{N}(0, 1))$. For linear regression, we would need the following generalization:

**Testing Problem 2.8** (Conditional NGCA)**.** Let $\{\mathcal{H}_y\}_{y \in \mathbb{R}}$ be a family of univariate distributions and $R$ be a univariate distribution. Consider Testing Problem 2.2 over $(x, y)$ on the domain $\mathbb{R}^d \times \mathbb{R}$ with

- ("Null") Under $\mathcal{P}$: $x \sim \mathcal{N}(0, \mathbf{I}_d)$ and $y \sim R$ independently.
- ("Alternate") Under $\mathcal{Q}_v$: $y \sim R$ and conditioned on $y = y_0$, $X|_{y=y_0} \sim P_v^{\mathcal{H}_{y_0}}$.

Building on [DKS17], [DKS19] showed SQ-hardness for the problem above if (i) $\mathcal{H}_y$ matches moments with $\mathcal{N}(0, 1)$ for (nearly) all $y \in \mathbb{R}$ and (ii) $\chi^2(\mathcal{Q}_v, \mathcal{P}) < \infty$. Unfortunately, neither of these conditions holds for us, and we need more flexible and powerful tools to bypass these limitations.

**Discrete Gaussian** We define Discrete Gaussian distributions that are central in our analysis.

**Definition 2.9.** For a center $\mu \in \mathbb{R}$, deviation $\sigma > 0$, base $\theta$, and spacing $s > 0$, define the distribution $\mathsf{DG}'[\mu, \sigma, \theta, s]$ to be the positive measure over $\theta + s\mathbb{Z}$ that assigns mass $s\phi_{\mu,\sigma}(\theta + si)$ for all $i \in \mathbb{Z}$; here, $\phi_{\mu,\sigma}$ denotes the pdf of the Gaussian distribution with mean $\mu$ and standard deviation $\sigma$. We use $\mathsf{DG}[\mu, \sigma, \theta, s]$ to denote the normalized probability distribution.

Discrete Gaussians behave similarly to Gaussians with respect to low-degree polynomials:

**Fact 2.10** ([DKRS23, Fact C.3] and [DK22, Lemma 3.12])**.** For any polynomial $p$ of degree at most $k$ and $\theta \in \mathbb{R}$ and $s > 0$, we have that $\left|\mathbb{E}_{G \sim \mathcal{N}(0,1)}[p(G)] - \mathbb{E}_{Y \sim \mathsf{DG}[0,1,\theta,s]}[p(Y)]\right| \lesssim \sqrt{\mathbb{E}_{G \sim \mathcal{N}(0,1)}[p^2(G)]}k!2^{O(k)}\exp(-\Omega(1/s^2))$.

# 3 Proof of Theorem 1.6

In this section, we will prove our main result Theorem 1.6. The first step is to make a judicious choice of the noise distribution $E$. For reasons outlined in Section 1.2, we choose $E$ to be a discrete Gaussian with $\sigma^2 \approx 1$ and spacing $s$ (eventually set to $\widetilde{\Theta}(\alpha)$).

As the second step, we note that the resulting testing problem is an instance of conditional NGCA.

**Testing Problem 3.1** (NGCA with Discrete Gaussian). For $y \in \mathbb{R}$, define the distribution $A_y := \mathsf{DG}\big[\mu_y, \widetilde{\sigma}, \theta_y, s'\big]$ with parameter values in Definition 3.2. Consider Testing Problem 2.8 with

- (Marginal of $y$) $R := R^*_{\rho,E}$ with $E = \mathsf{DG}\big[0, \sigma, 0, s\big]$.[3]
- (Conditional NGCA) For each $y \in \mathbb{R}$, $\mathcal{H}_y$ is equal to $A_y$.

We denote the corresponding null by $P$ and the alternate for direction $v$ by $Q_v$.

We mention the parameter choices that we shall enforce from now on:

---

**Definition 3.2.** Let signal strength $\rho \in (0, \rho_0)$ for sufficiently small $\rho_0 > 0$, standard deviation $\sigma \in (0.5, 1)$, spacing $s \in (0, 1)$ satisfy the following values:

- $\sigma = \sqrt{1 - \rho^2}$,
- $s' = s/\rho \leq 0.001$,

- $\mu_y := \rho \, y$,
- $\theta_y = y/\rho$.

---

That is, for each $y$, the conditional distribution of the covariates in the hidden direction is a discrete Gaussian with mean $\mu_y$ (scaling linearly with $y$) and standard deviation $\sigma$ (slightly smaller than 1). While these parameters might look a bit obscure, they perfectly resemble the typical setting of $E = \mathcal{N}(0, \sigma^2)$.[4] The next result, proved in Appendix B.1, shows that Testing Problem 1.3 is equivalent to Testing Problem 3.1.

**Proposition 3.3.** *Testing Problem 3.1 is equivalent to Testing Problem 1.3 when $E = \mathsf{DG}\big[0, \sigma, 0, s\big]$.*

Thus, to prove Theorem 1.6, it suffices to consider Testing Problem 3.1, which is a conditional NGCA instance. Since the distributions $\{A_y\}_{y \in \mathbb{R}}$ are (necessarily) degenerate, the lower bound machinery of SDA and pairwise correlations developed in [DKS17; DKS19] for (conditional) NGCA lead only to vacuous bounds. To bypass this degeneracy, we will instead use Proposition 2.6 (C.II) and will show that for any bounded query $f : \mathcal{U} \to [0, 1]$,

$$\mathbb{P}_{v \sim \mathcal{S}^{d-1}}\big\{|\mathbb{E}_{Z \sim Q_v}[f(Z)] - \mathbb{E}_{Z \sim P}[f(Z)]| \geq \text{``large''}\big\} \leq \text{``tiny''}, \tag{1}$$

where the notion of being "large" is according to Definition 2.5 for $m = \widetilde{o}(\rho^2/\sqrt{d})$. However, it is unwieldy to compute (or upper bound) this probability. Hence, we first take a detour to a related testing problem with the more usual continuous Gaussian noise in the next section.

## 3.1 Conditional NGCA with Continuous Gaussian

As mentioned in the introduction, we use the similarity of discrete Gaussian with continuous Gaussian (with respect to polynomials) as an analysis tool. We define the analogous testing problem with continuous Gaussian noise below.

**Testing Problem 3.4.** For $y \in \mathbb{R}$, let $B_y$ denote the distribution $\mathcal{N}\big(\mu_y, \sigma^2\big)$ with parameters as in Definition 3.2. Consider Testing Problem 2.8 with

- (Marginal of $y$) $R := R^*_{\rho,E}$ with $E = \mathsf{DG}\big[0, \sigma, 0, s\big]$.
- (Conditional NGCA) For each $y \in \mathbb{R}$, $\mathcal{H}_y$ is equal to $B_y$.

We denote the corresponding null by $P$ (same as 3.1) and the alternate for direction $v$ by $T_v$.

**Remark 3.5.** Observe that the alternate above $T_v$ does not correspond to the following (Gaussian) linear model: $y = \rho v^\top x + z$ for $x \sim \mathcal{N}(0, \mathbf{I}_d)$ and $z \sim \mathcal{N}(0, \sigma^2)$ independently of $x$. This is because the marginal of $Y$ under the aforementioned linear model would have been Gaussian $\mathcal{N}(0, 1)$, while it is $R$ in Testing Problem 3.4 (which is not Gaussian).

Before establishing similarity with discrete Gaussian quantitatively, we first establish that Testing Problem 3.4 is SQ-hard. In fact, we show the stronger result that the associated SDA is large.

**Proposition 3.6** (SQ Hardness of Continuous Noise). *Consider Testing Problem 3.4. Then for any $m \in \mathbb{N}$ and $q \in \mathbb{N}$ satisfying $\frac{\rho^2 \sqrt{\log(1/q)}}{\sqrt{d}} \lesssim \frac{1}{m}$, we have that $\mathrm{SDA}(m) \gtrsim q$.*

---

[3]Recall that $R^*_{\rho,E}$ is the distribution of $x + z$ for $x \sim \mathcal{N}(0, \rho^2)$ and $z \sim E$ independent of each other.

[4]The conditional distribution of $x|y$ in the hidden direction $v$ would be $\mathcal{N}(\mu_y, \sigma^2)$ [DKPPS21, Fact 3.3].

*Proof Sketch.* We prove a bound on SDA by calculating an analytic upper bound on the pairwise correlation $\chi_P(T_v, T_{v'})$. Since the marginal of $y$ is identical ($R$) under $P, T_v$, and $T_{v'}$, the pairwise correlation is equal to $\mathbb{E}_{y \sim R}[\chi_{\mathcal{N}(0, \mathbf{I}_d)}(P_v^{B_y}, P_{v'}^{B_y})]$. Since $P_v^{B_y}$ and $P_v^{B_y}$ are Gaussians, there is a closed-form expression (in terms of $y$), which we integrate out using nice properties of $R$. $\square$

As a consequence, Proposition 2.6 (C.I) implies that for any $f : \mathcal{U} \to [0, 1]$ and $m = o(\frac{\rho^2 \log^4 d}{\sqrt{d}})$,

$$\mathbb{P}_{v \sim \mathcal{S}^{d-1}}\{|\mathbb{E}_{T_v}[f] - \mathbb{E}_P[f]| \geq \text{"threshold of VSTAT}(m)\text{"}\} \leq \frac{1}{d^{\omega(\log^2 d)}}. \tag{2}$$

## 3.2 Hardness of Distinguishing Discrete and Gaussian Noise

Towards establishing (1), a natural step after proving (2) is to argue that, with high probability, $|\mathbb{E}_{T_v}[f(Z)] - \mathbb{E}_{Q_v}[f(Z)]|$ is small. This is exactly what we establish in the next result, which is our main technical result:

**Theorem 3.7.** *Suppose that (i) $\alpha \gg \frac{1}{d^{\mathrm{polylog}(d)}}$ and $\rho^2 \geq s^2 \log^C(d/\alpha)$ for a large constant $C > 0$. Then for any $f : \mathcal{U} \to [0, 1]$, it holds that $\mathbb{P}_{v \sim \mathcal{S}^{d-1}}\left[\left|\mathbb{E}_{Q_v}[f] - \mathbb{E}_{T_v}[f]\right| \gtrsim \left(\frac{\alpha}{d}\right)^{\log^2(d/\alpha)}\right] \leq \frac{1}{d^{\log^2 d}}$.*

In the remainder of this section, we detail the proofs and intuition for the above result.

As a first step, we do a Hermite expansion of the function $f$ as in [DKRS23], but generalized to the setting of conditional NGCA. However, for technical reasons due to the degeneracy of $A_y$ and hence $Q_v$, we would need to perform another truncation operation.

**Definition 3.8.** Define $\widetilde{A}_y$ to be the univariate distribution $A_y$ conditioned on $\{z : |z| \leq d\}$ and let $\widetilde{Q}_v$ to be analogous to $Q_v$ but with $\widetilde{A}_y$ instead of $A_y$.

We now use the Hermite expansion to obtain the following result:

**Proposition 3.9.** *Let $f : \mathbb{R}^d \times \mathbb{R} \to [0, 1]$. For any $L \leq [1, \frac{d}{2}]$, $\ell \in \mathbb{N}$, and $\widetilde{f} := f \cdot \mathbb{1}_{|y| \leq L}$, we have*

$$|\mathbb{E}_{T_v}[f] - \mathbb{E}_{Q_v}[f]| \lesssim e^{-\Omega(L^2)} + e^{-\Omega(d)} + \sum_{k=1}^{\ell} \max_{|y| \leq L} \left|\widetilde{\mathbf{A}}_{k,y} - \mathbf{B}_{k,y}\right| \cdot \mathbb{E}_{y \sim R}\left[\left|\langle v^{\otimes k}, \mathbf{T}_{k,y}\rangle\right|\right]$$
$$+ \left|\mathbb{E}_{\widetilde{Q}_v}[\widetilde{f}^{>\ell}] - \mathbb{E}_{Q_v}[\widetilde{f}^{>\ell}]\right| \tag{3}$$

*where $\mathbf{T}_{k,y} := \mathbb{E}_{x \sim \mathcal{N}(0, \mathbf{I}_d)}[\widetilde{f}_y(x) H_k(x)]$, $\widetilde{\mathbf{A}}_{k,y} := \mathbb{E}_{x \sim \widetilde{A}_y}[h_k(x)]$, and $\mathbf{B}_{k,y} := \mathbb{E}_{x \sim B_y}[\widetilde{f}(x)]$.*

*Proof Sketch.* Since $R_{\rho, E}$ has very light tails, we can replace $f$ with $\widetilde{f}$ which leads to a difference of at most $\mathbb{P}(|y| \geq L) \lesssim e^{-\Omega(L^2)}$.
Next, we decompose $\widetilde{f}$ as $\widetilde{f}^{\leq \ell}$ and $\widetilde{f}^{>\ell}$, where the $\widetilde{f}^{>\ell}$ term appears as is in (3) and can be ignored momentarily. Then, using law of total expectation, we can write $\mathbb{E}_{(x,y) \sim TQ_v}[\widetilde{f}^{\leq \ell}] = \mathbb{E}_y[\mathbb{E}_x[\widetilde{f}_y^{\leq \ell}(x)]]$. The result in [DKRS23, Lemma 3.3] implies that $\mathbb{E}_{P_v^{A_y}}[\widetilde{f}_y^{\leq \ell}(x)] = \sum_{k=0}^{\ell} \mathbf{A}_{k,y} \langle v^{\otimes k}, \mathbf{T}_{k,y}\rangle$ for $\mathbf{A}_{k,y} := \mathbb{E}_{x \sim A_y}[h_k(x)]$. A similar argument holds for $B_y$. Taking the difference and integrating over $y$, we obtain $\mathbb{E}_{Q_v}[\widetilde{f}^{\leq \ell}] - \mathbb{E}_{T_v}[\widetilde{f}^{\leq \ell}] = E_y\left[\sum_{k=1}^{\ell} \left(\mathbf{A}'_{k,y} - \mathbf{B}_{k,y}\right) \langle v^{\otimes k}, \mathbf{T}_{k,y}\rangle\right]$.

Since $\widetilde{f}$ is zero for $|y| \geq L$, $\mathbf{T}_{k,y}$ is also zero for large $y$ and we can take the maximum only over $|y| \leq L$, yielding (3) roughly. However, later on, we would still need to control $\mathbb{E}_Q[\widetilde{f}^{>\ell}]$, which could potentially be large because of degeneracy and unboundedness of $A_y$s. Therefore, we replace $A_y$s with $\widetilde{Q}_y$s to make it bounded; using concentration of $A_y$s, this leads to an additional $e^{-\Omega(d^2)}$ term. $\square$

Thus, we crucially need to control $|\widetilde{\mathbf{A}}_{k,y} - \mathbf{B}_{k,y}|$ and obtain high-probability estimates (over randomness in $v$) on $|\langle v^{\otimes k}, \mathbf{T}_{k,y}\rangle|$. We begin with the former, whose proof is deferred to Appendix B.5; we note that the key ingredient in proving this result is Fact 2.10.

**Lemma 3.10** (Closeness of Hermite Coefficients). *For any $y \in \mathbb{R}$ and $k \in \mathbb{N}$, we have*

- *(Tighter for small $k$) $|\widetilde{\mathbf{A}}_{k,y} - \mathbf{B}_{k,y}| \lesssim \max\left(1, |\mu_y|^k\right) k^{O(k)} \cdot \left(e^{-\Omega\left(\frac{\rho^2}{s^2}\right)} + e^{-\Omega(d)}\right)$.*
- *(Tighter for larger $k$) $|\widetilde{\mathbf{A}}_{k,y} - \mathbf{B}_{k,y}| \lesssim e^{O(\mu_y^2)}$.*

Thus, Lemma 3.10 implies that (i) for small $k$, the difference is inverse super-polynomially small if $\rho^2 \gg s^2 \mathrm{polylog}(d) \asymp \alpha^2 \mathrm{polylog}(d)$ and (ii) it stays bounded by $O(1)e^{L^2}$ for $|y| \le L$ for any $k$.

We now turn to computing high-probability estimates on $\mathbb{E}_y|\langle v^{\otimes k}, \mathbf{T}_{k,y}\rangle|$. Here, we reparameterize the arguments in [DKRS23] and obtain the following result, whose proof is deferred to Appendix B.6.

**Proposition 3.11.** *Let* $\{\mathbf{T}_{k,y}\}_{k\in\mathbb{N},y\in\mathbb{R}}$ *be tensors with* $\|\mathbf{T}_{k,y}\|_2 \le 1$ *for all* $k \in \mathbb{N}, y \in \mathbb{R}$, *and let* $t \in \mathbb{N}$ *be arbitrary. Then for any* $\delta \in (0,1)$, *it holds with probability* $1-\delta$ *over a random unit vector* $v$ *that*

$$\mathbb{E}_y\Big[\sum_{k=1}^{t} |\langle v^{\otimes k}, \mathbf{T}_{k,y}\rangle|\Big] \lesssim t \quad and \quad \sum_{k>t+1}^{\infty} \mathbb{E}_y\big[|\langle v^{\otimes k}, \mathbf{T}_k\rangle|\big] \lesssim d^{O(1)}\Big(\frac{t\log\frac{t}{\delta}}{d}\Big)^{t/4} + d^{O(1)} \cdot \frac{1}{\delta}e^{-\frac{Cd}{\log\frac{d}{\delta}}} \, .$$

The result above is applicable to our setting because for each $y \in \mathbb{R}$: $\sum_{k=1}^{\infty} \|\mathbf{T}_{k,y}\|_2^2 = \|\widetilde{f}_y\|_{L_2(\mathcal{N}(0,\mathbf{I}_d))}^2 \le 1$, where the equality uses the orthonormality of Hermite tensors under $\mathcal{N}(0,\mathbf{I}_d)$ and the inequality uses that $\widetilde{f}$ is bounded by 1.

### 3.2.1 Proof sketch of Theorem 3.7

We are now ready to present a proof sketch of Theorem 3.7. Combining Proposition 3.9 with Lemma 3.10 and Proposition 3.11 and the fact that $|\mu_y| \le L$ for $L \ge 1$, we obtain that for any $t \in \mathbb{N}$ with probability at least $1 - d^{-\log^2 d}$,

$$\big|\mathbb{E}_{T_v}[f] - \mathbb{E}_{Q_v}[f]\big| \lesssim e^{-\Omega(L^2)} + e^{-\Omega(d)} + L^t t^{O(t)}\big(e^{-\Omega(\rho^2/s^2)} + e^{-\Omega(d)}\big) t$$
$$+ e^{cL^2}(dt)^{O(1)}\Big(\frac{t\log t \log^3 d}{d}\Big)^{t/4} + e^{cL^2} e^{-d/\mathrm{polylog}(d)} + \big|\mathbb{E}_{T_v}[\widetilde{f}^{>\ell}] - \mathbb{E}_{\widetilde{Q}_v}[\widetilde{f}^{>\ell}]\big|.$$

For $L = \log^5 d$, $t = L^6$ and $\rho = st^2$, the sum of the first four terms is at most $O(e^{-L^2}) \le d^{-\log^2(d/\alpha)}$. For the last term, we show that taking $\ell$ large enough suffices—this argument uses Fact 2.1 and the truncation of $A_y$ as per [DKRS23]; see Appendix B.8 for details.

## 3.3 Proof Sketch of Theorem 1.6

Since $E = \mathrm{DG}\big[0, \sigma, 0, s\big]$, we have that $\mathbb{P}_E(z = 0) = \Theta(s/\sigma)$ up to normalization $1 \pm e^{-1/s^2}$; see Fact 2.10. Taking $s = \Theta(\alpha)$, we get that $\mathbb{P}(z = 0) \ge \alpha$ satisfying our model. To establish SQ lower bound, it suffices to show that the probability of success of $f$ on distinguishing $Q_v$ and $P$ with $m \ll m_0 := \widetilde{\Theta}(\sqrt{d}/\alpha^2)$ accuracy complexity is at most $1/q_0 := d^{-\Omega(\log^2 d)}$ (cf. Proposition 2.6). Let this event be $\mathcal{E}_{f,v,m}$. We now define the following events:

- $\mathcal{E}'_{f,v,m} := \{v : \big|\mathbb{E}_{Q_v}[f(x,y)] - \mathbb{E}_{T_v}[f(x,y)]\big| \ge \frac{1}{4m^2}\}$; Theorem 3.7 implies that $\mathbb{P}(\mathcal{E}'_{f,v,m}) \le \frac{1}{2q_0}$.
- $\mathcal{E}''_{f,v,m}$ is defined analogous to $\mathcal{E}_{f,v,m}$ but with (i) $T$ instead of $Q$ and (ii) $Cm$ accuracy complexity as opposed to $m$ for a large constant $C$. Proposition 3.6 implies $\mathbb{P}(\mathcal{E}''_{f,v,m}) \le \frac{1}{2q_0}$.

Since $\mathcal{E}_{f,v} \subset \mathcal{E}'_{f,v,m} \cup \mathcal{E}''_{f,v,m}$ (Claim B.7), the desired result follows by a union bound.

# 4 Conclusions and Open Problems

In this work, we studied the fundamental problem of noiseless linear regression under Gaussian marginals with additive oblivious contamination. Our main result is an information-computation tradeoff for SQ algorithms, suggesting that efficient learners require sample complexity at least quadratic in $1/\alpha$, where $\alpha$ is the fraction of inliers, while linear dependence in $1/\alpha$ information-theoretically suffices. An immediate open problem concerns the dependence on $d$ in the lower bound. Specifically, it is a plausible conjecture that there exists a lower bound of $\Omega(d/\alpha^2)$ on the computational sample complexity of the problem (thus, exactly matching the sample complexity of known algorithms). We note that such a lower bound would require a new hardness construction, as our hard testing instance is efficiently solvable with $O(d^{1/2}/\alpha^2)$ samples.

## Acknowledgements

ID was supported by NSF Medium Award CCF-2107079 and an H.I. Romnes Faculty Fellowship. CG was supported by NSF Grants ECCS-2216912 and DMS-2310769 and an Alfred Sloan fellowship. DK was supported by NSF Medium Award CCF-2107547. AP was supported by Research Pod on Resilience in Brain, Natural, and Algorithmic Systems at the Simons Institute, UC Berkeley.

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

# Supplementary Material

The Appendix is organized as follows: Appendix A contains additional preliminaries and background on SQ algorithms. Appendix B contains proofs deferred from Section 3. Appendix C gives a computationally-efficient reduction from testing to estimation. Appendix D gives an inefficient SQ algorithm that uses VSTAT oracle with accuracy complexity linear in $\frac{1}{\alpha}$, whereas Appendix E gives an efficient SQ algorithm that uses a VSTAT oracle with accuracy complexity quadratic in $\frac{1}{\alpha}$.

## A   Additional Preliminaries

We say a random variable $X$ or a distribution $P$ is $\sigma$-subgaussian if $\mathbb{P}(|X| \geq t) \leq 2\exp(-ct^2/\sigma^2)$ for all $t > 0$; here $c$ is an absolute constant.

**Fact A.1.** There exists a finite constant $a_0 > 0$ such that if $X$ is $\sigma$-subgaussian then $\left|\mathbb{E}\left[e^{a\frac{X^2}{\sigma^2}}\right] - 1\right| \lesssim |a|$ for $|a| \leq a_0$.

*Proof.* We use expansion of $e^x$ and the fact that $\mathbb{E}[|X|^p] \leq (C\sigma\sqrt{p})^p$ for a $\sigma$-subgaussian distribution [Ver18, Proposition 2.5.2] to get

$$\mathbb{E}[e^{\frac{aX^2}{\sigma^2}} - 1] = \mathbb{E}\left[\sum_{i=1}^{\infty} a^i \frac{X^{2i}}{\sigma^{2i}i!}\right] \leq \sum_{i=1}^{\infty} a^i \frac{(c\sigma\sqrt{2i})^{2i}}{\sigma^{2i}i!} \leq \sum_{i=1}^{\infty} \frac{(\sqrt{e}ac\sqrt{2i})^{2i}}{i^i} \leq \sum_{i=1}^{\infty} (\sqrt{e}ac\sqrt{2})^{2i},$$

which is of order $O(a)$ for small enough $a$ because it then converges as a geometric sequence. □

For completeness, we provide the proof of Fact 2.1.

**Fact 2.1.** For every function $f : \mathbb{R}^d \to [-1, 1]$, $\|f^{>\ell}\|_{L_2(\mathcal{N}(0,\mathbf{I}_d))} \to 0$ as $\ell \to \infty$. Furthermore, for all $f : \mathbb{R}^d \times \mathbb{R} \to [-1, 1]$ and univariate measures $R$, $\|f^{>\ell}\|_{L_2(\mathcal{N}(0,\mathbf{I}_d)\times R)} \to 0$ as $\ell \to \infty$.

*Proof.* The first statement is a simple consequence of the fact that Hermite polynomials are a complete orthonormal system of $L^2(\mathbb{R}^d, \mathcal{N}(0, \mathbf{I}_d))$.

For the second statement, we shall use dominated convergence theorem. Define the residue $f_y^{>\ell}(x) := f(x,y) - f_y(x)^{>\ell}$ and $J_\ell(y) := \|f_y^{>\ell}\|_{L_2(\mathcal{N}(0,\mathbf{I}_d))}^2$. Observe that $\mathbb{E}_{y\sim\mathbb{R}}[J_\ell(y)] = \|f^{>\ell}\|_{L_2(\mathcal{N}(0,\mathbf{I}_d)\times R)}$. The first statement implies that for each $y \in \mathbb{R}$, $J_\ell(\cdot) \to 0$ as $\ell \to \infty$. Furthermore, $J_\ell$ is uniformly bounded by 4 as follows:

$$J_\ell(y) = \|f_y(x) - f_y^\ell(x)\|_{L_2(\mathcal{N}(0,\mathbf{I}_d))}^2 \leq 2\|f_y\|_{L^2(\mathcal{N}(0,\mathbf{I}_d))}^2 + 2\|f_y^{\leq\ell}\|_{L^2(\mathcal{N}(0,\mathbf{I}_d))}^2 \leq 4,$$

where we use Parseval's identity to say $\|f_y^{\leq\ell}\|_{L^2(\mathcal{N}(0,\mathbf{I}_d))}^2 \leq \|f_y\|_{L^2(\mathcal{N}(0,\mathbf{I}_d))}^2$ and that $|f_y| \leq 1$. Since $J_\ell \to 0$ pointwise as $\ell \to \infty$ and $0 \leq J_\ell \leq 4$ uniformly, by the dominated convergence theorem, $\mathbb{E}_y[J_\ell(y)] \to 0$ as $\ell \to \infty$.

□

### A.1   Statistical Query Algorithms

Instead of getting sample access, SQ algorithms interact with the underlying distribution through an oracle. Observe that there are many ways of implementing a VSTAT$(m)$ oracle, especially when the SQ algorithm $\mathcal{A}$ makes multiple requires—all we require is that each response is a valid VSTAT$(m)$ response to each query.

Recall the notion of success from Definition 2.5. The notion of success is intimately tied to the SDA as shown by the following result:

**Proposition A.2** (SQ lower bounds using SDA; Proposition 2.6 (C.I)). *For any query $f : \mathcal{Z} \to [0, 1]$, the following holds:* $\mathbb{P}_{v\sim\mathcal{S}^{d-1}}\left(\mathcal{E}_{f,v,m}\right) \leq \frac{1}{\mathrm{SDA}(7m)}$.

We use the arguments implicit in [FGRVX17; BBHLS21; DKRS23].

*Proof.* Here, we assume that $\mathcal{Q}_v$ has a valid density with respect to $\mathcal{P}$. For a $v \in \mathcal{S}^{d-1}$ and $z \in \mathcal{Z}$, we use $q_v(z)$ to denote the Radon–Nikodym derivative of $\mathcal{Q}_v$ with respect to $\mathcal{P}$. Observe that

$$\mathbb{E}_{\mathcal{Q}_v}[f] - \mathbb{E}_{\mathcal{P}}[f] = \langle q_v - 1, \ f \rangle_{\mathcal{P}}.$$

Fix a query $f$ and assume that $a_1 := \mathbb{E}_{\mathcal{P}}[f] \leq \frac{1}{2}$, otherwise apply the following arguments to $1 - f$. We shall show that $\mathbb{P}(\mathcal{E}_{f,v,m}) \leq 1/\mathrm{SDA}(7m)$ by contradiction. Suppose $\mathbb{P}(\mathcal{E}_{f,v,m}) > 1/\mathrm{SDA}(7m)$.

Lemma 3.5 in [FGRVX17] implies that for any $m \geq 1, 0 \leq a_1, a_2 \leq 1$,

$$\text{if} \quad |a_1 - a_2| \ \geq \ \max\!\left(\tfrac{1}{m}, \ \min\!\left(\sqrt{\tfrac{a_1(1-a_1)}{m}}, \ \sqrt{\tfrac{a_2(1-a_2)}{m}}\right)\right), \quad \text{then} \quad |a_1 - a_2| \ \geq \ \sqrt{\tfrac{a_1(1-a_1)}{3m}}.$$

Applying this with $a_1 = \mathbb{E}_{\mathcal{P}}[f]$ and $a_2 = \mathbb{E}_{\mathcal{Q}_v}[f]$, then if $f$ succeeds on $v$, then

$$\mathbb{1}_{\mathcal{E}_{f,v,m}} \left| \langle q_v - 1, f \rangle_{\mathcal{P}} \right| \ \geq \ \sqrt{\tfrac{a_1}{6m}} \, \mathbb{1}_{\mathcal{E}_{f,v,m}}.$$

Taking expectation over $v$ and squaring gives

$$
\begin{aligned}
\mathbb{P}(\mathcal{E}_{f,v,m})^2 \cdot \frac{a_1}{6m} &\leq \left( \mathbb{E}_v \left[ \mathbb{1}_{\mathcal{E}_{f,v,m}} \left| \langle q_v - 1, f \rangle_{\mathcal{P}} \right| \right] \right)^2 \\
&= \left( \mathbb{E}_v \left[ \mathbb{1}_{\mathcal{E}_{f,v,m}} \langle q_v - 1, f \rangle_{\mathcal{P}} \cdot \mathrm{sgn}_{\langle q_v - 1, f \rangle_{\mathcal{P}}} \right] \right)^2 \\
&= \left( \mathbb{E}_v \left[ \mathbb{E}_{\mathcal{P}} \left[ \mathbb{1}_{\mathcal{E}_{f,v,m}} \left( q_v(Z) - 1 \right) \left( f(Z) \right) \cdot \mathrm{sgn}_{\langle q_v - 1, f \rangle_{\mathcal{P}}} \right] \right] \right)^2 \\
&= \left( \mathbb{E}_{\mathcal{P}} \left[ \mathbb{E}_v \left[ \mathbb{1}_{\mathcal{E}_{f,v,m}} \left( q_v(Z) - 1 \right) \left( f(Z) \right) \cdot \mathrm{sgn}_{\langle q_v - 1, f \rangle_{\mathcal{P}}} \right] \right] \right)^2 \\
&= \left\langle \mathbb{E}_v \left[ \mathbb{1}_{\mathcal{E}_{f,v,m}} \left( q_v - 1 \right) \mathrm{sgn}_{\langle q_v - 1, f \rangle_{\mathcal{P}}} \right], f \right\rangle_{\mathcal{P}}^2 \\
&\leq \|f\|_{L_2(\mathcal{P})}^2 \cdot \left\| \mathbb{E}_v \left[ \mathbb{1}_{\mathcal{E}_{f,v,m}} \left( q_v - 1 \right) \mathrm{sgn}_{\langle q_v - 1, f \rangle_{\mathcal{P}}} \right] \right\|_{L_2(\mathcal{P})}^2 \\
&= a_1 \cdot \left( \mathbb{E}_{v,v',\mathcal{P}} \left[ \mathbb{1}_{\mathcal{E}_{f,v,m}} \mathbb{1}_{\mathcal{E}_{f,v',m}} \left( q_v(Z) - 1 \right) \left( q_{v'}(Z) - 1 \right) \mathrm{sgn}_{\langle q_v - 1, f \rangle_{\mathcal{P}}} \mathrm{sgn}_{\langle q_{v'} - 1, f \rangle_{\mathcal{P}}} \right] \right) \\
&= a_1 \cdot \left( \mathbb{E}_{v,v'} \left[ \mathbb{1}_{\mathcal{E}_{f,v,m}} \mathbb{1}_{\mathcal{E}_{f,v',m}} \langle q_v - 1, q_{v'} - 1 \rangle_{\mathcal{P}} \, \mathrm{sgn}_{\langle q_v - 1, f \rangle_{\mathcal{P}}} \mathrm{sgn}_{\langle q_{v'} - 1, f \rangle_{\mathcal{P}}} \right] \right) \\
&\leq a_1 \cdot \left( \mathbb{E}_{v,v'} \left[ \mathbb{1}_{\mathcal{E}_{f,v,m}} \mathbb{1}_{\mathcal{E}_{f,v',m}} \left| \langle q_v - 1, q_{v'} - 1 \rangle_{\mathcal{P}} \right| \right] \right).
\end{aligned}
$$

Dividing both sides by $a_1 \mathbb{P}(\mathcal{E}_{f,v,m})^2$ and noting independence gives, for $\mathcal{E} = \mathcal{E}_{f,v,m} \cap \mathcal{E}_{f,v',m}$,

$$\frac{1}{6m} \ \leq \ \mathbb{E}_{v,v'} \left[ \left| \langle q_v - 1, \ q_{v'} - 1 \rangle_{\mathcal{P}} \right| \ \Big| \ \mathcal{E} \right].$$

Since $\mathbb{P}(\mathcal{E}) = \mathbb{P}(\mathcal{E}_{f,v,m})^2 \geq \frac{1}{\mathrm{SDA}(7m)^2}$, the definition of SDA implies the RHS is $< \frac{1}{7m}$, a contradiction. Hence, $\mathbb{P}(\mathcal{E}_{f,v,m}) \leq \frac{1}{\mathrm{SDA}(7m)}$. $\qquad\square$

**Proposition A.3** (Query Complexity Lower Bound; Proposition 2.6 (C.II)). *Fix a $m \in \mathbb{N}$ and $q \in \mathbb{N}$. Suppose that for all bounded queries $f : \mathcal{Z} \to [0,1]$, the probability of success is small as follows:*

$$\mathbb{P}_{v \sim \mathcal{S}^{d-1}} \left( \mathcal{E}_{f,v,m} \right) \leq \frac{1}{q}. \tag{4}$$

*Then any (potentially randomized and adaptive to the responses of the previous queries) SQ algorithm $\mathcal{A}$ for solving Testing Problem 2.2 (with failure probability less than $0.25$) must use either $\Omega(q) = \Omega(\mathrm{SDA}(7m))$ queries or at least one query as powerful as VSTAT$(m+1)$.*

Again, we use ideas implicit in [FGRVX17; BBHLS21; DKRS23].

*Proof.* We will fix the VSTAT oracle to be a deterministic oracle $\mathbf{V}^*$ defined below (independent of the algorithm $\mathcal{A}$). Since the oracle is fixed, it suffices to show lower bounds against deterministic algorithms.

Consider the following oracle $\mathbf{V}^*$:

- If $\Theta = \mathcal{P}$, then for any query $f$, it returns $\mathbb{E}_{\mathcal{P}}[f]$.

- If $\Theta = \mathcal{Q}_v$, then it answers differently based on the "niceness" of $f$:
  - ("a good query for $\mathbf{V}^*$ on $v$") If $\mathbb{E}_{\mathcal{P}}[f]$ is a valid VSTAT($m$) response, then answer $\mathbb{E}_{\mathcal{P}}[f]$.
  - (otherwise) Return $\mathbb{E}_{\mathcal{Q}_v}[f]$.

Observe that the oracle $\mathbf{V}$ above is a valid VSTAT($m$) oracle for all both null and all alternate.

Now, let $\mathcal{A}'$ be any deterministic SQ algorithm (deterministic as a function of the answers of the oracle) that solves the testing problem with queries $f_1, \ldots, f_{q'}$ for $q' = aq$ for some $a < 1$ to be decided soon.

Consider the case when $\Theta = P$. Recall that the adversary returns $\mathbb{E}_{\mathcal{P}}[f_i]$ for $i \in [q']$, which is a valid response. Then, the accuracy guarantee of $\mathcal{A}$ implies that $\mathcal{A}$ must output "null" on these instances (because it is deterministic); otherwise the failure probability $\mathbb{P}(\widehat{\Gamma} \neq \Gamma) \geq 0.5$.

Now, consider the alternate case where $\Theta = \mathcal{Q}_v$ for $v \sim \mathcal{S}^{d-1}$. Observe that if $\mathcal{E}^{\complement}_{f,v,m}$ holds for a query $f$, then it is a "a good query for $\mathbf{V}^*$ on $v$". By assumption, the probability (over $v$) that any fixed query $f$ is not good is at most $\mathbb{P}_{\Theta}(\mathcal{E}_{f,v,m}) \leq 1/q$. Thus, by a union bound, the probability (over $v$) that all the queries $\{f_i\}_{i=1}^{q'}$ are good for $\mathbf{V}^*$ is at least $1 - a$. When all the queries are good, the algorithm's input is the same as in the null case, and hence the algorithm must answer "null". Therefore, the overall failure probability is at least $0.5(1 - a) \geq 0.25$, which is a contradiction. $\quad\square$

## A.2 Pairwise Correlation

We will use the following closed-form expression for the pairwise correlations between Gaussians:

**Lemma A.4.** *Let unit vectors $u, v \in \mathbb{R}^d$ and scalars $a \in \mathbb{R}$ and $\gamma \in (0, 1)$. Let $\cos\theta = u^\top v$. Then*

$$1 + \chi_{\mathcal{N}(0, \mathbf{I}_d)}\left(\mathcal{N}(av, I - \gamma vv^\top), \mathcal{N}(au, I - \gamma uu^\top)\right) = \frac{\exp\left(\frac{\alpha^2 \cos\theta}{1 + \gamma \cos\theta}\right)}{\sqrt{1 - \gamma^2 \cos^2\theta}}.$$

*Proof.* For any two Gaussians $A = \mathcal{N}(\mu_1, \Sigma_1)$ and $B = \mathcal{N}(\mu_2, \Sigma_2)$, the average correlation with respect to the standard Gaussian can be calculated as follows:

$$\chi_{\mathcal{N}(0, \mathbf{I}_d)}(A, B) = \frac{\exp((h' - h)/2)}{\sqrt{s_1 s_2}\sqrt{s_{1,2}}} \qquad \text{where} \tag{5}$$

- $A = (\Sigma_1^{-1} + \Sigma_2^{-1} - I)^{-1}$ • $s_1 = \det(\Sigma_1)$ • $s_2 = \det(\Sigma_1)$ • $s_{1,2} = \det(A^{-1})$
- $h = \mu_1^\top \Sigma_1^{-1} \mu_1 + \mu_2^\top \Sigma_2^{-1} \mu_2$ • $y = \Sigma_1^{-1}\mu_1 + \Sigma_2^{-1}\mu_2$ • $h' = y^\top Ay$.

We will now instantiate the formula above in our context below.

- (Calculating $s_1$ and $s_2$) $s_1 = s_2 = 1 - \gamma$. Also define $b := 1 - \gamma$.

- (Calculating $h$) Moreover, $\Sigma_1^{-1} = (I - vv^\top) + b^{-1}vv^\top$ and $\Sigma_2^{-1} = (I - uu^\top) + b^{-1}uu^\top$. Therefore, $h = \mu_1^\top \Sigma_1^{-1} \mu_1 + \mu_1^\top \Sigma_1^{-1} \mu_1 = 2a^2 b^{-1}$.

- (Calculating $y$) The same calculations as above give $y = \Sigma_1^{-1}\mu_1 + \Sigma_1^{-1}\mu_1 = ab^{-1}(v + u)$.

- (Calculating $s_{1,2}$) We begin by calculating $A^{-1}$:
$$A^{-1} = \Sigma_1^{-1} + \Sigma_2^{-1} - I = (I - vv^\top) + b^{-1}vv^\top + (I - uu^\top) + b^{-1}uu^\top - I$$
$$= I + \frac{\gamma}{1 - \gamma} \cdot (vv^\top + uu^\top).$$

  Therefore, the determinant of $A^{-1}$ is $\frac{1 - \gamma^2 \cos^2\theta}{(1 - \gamma)^2}$ for $\alpha = \frac{\gamma}{1 - \gamma}$. This can be seen as follows by considering $2 \times 2$ matrices:
$$\det\left(I + \begin{bmatrix} \alpha & \alpha u^\top v \\ \alpha u^\top v & \alpha \end{bmatrix}\right) = \det\left(\begin{bmatrix} 1 + \alpha & \alpha u^\top v \\ \alpha u^\top v & 1 + \alpha \end{bmatrix}\right) = (1 + \alpha)^2 - \alpha^2 (u^\top v)^2,$$
  which equals the expression above.

- (Calculating $A$ and $h'$) Letting $U = [u; v] \in \mathbb{R}^{d\times 2}$ and $C = \alpha \mathbf{I}_2$, then

$$A = (I + UCU^\top)^{-1} = I - U(C^{-1} + U^\top U)U^\top$$

$$= I - U \left( \begin{bmatrix} \alpha^{-1} & 0 \\ 0 & \alpha^{-1} \end{bmatrix} + \begin{bmatrix} 1 & \cos\theta \\ \cos\theta & 1 \end{bmatrix} \right)^{-1} U^\top$$

$$= I - U \left( \begin{bmatrix} 1 + \alpha^{-1} & \cos\theta \\ \cos\theta & 1 + \alpha^{-1} \end{bmatrix} \right)^{-1} U^\top$$

$$= I - U \left( \begin{bmatrix} \gamma^{-1} & \cos\theta \\ \cos\theta & \gamma^{-1} \end{bmatrix} \right)^{-1} U^\top$$

$$= I - \frac{1}{1 - \gamma^2 \cos^2\theta} U \left( \begin{bmatrix} \gamma & -\gamma^2\cos\theta \\ \gamma^2\cos\theta & \gamma \end{bmatrix} \right)^{-1} U^\top$$

$$= I - \left( \frac{\gamma}{1 - \gamma^2 \cos^2\theta} \cdot (uu^\top + vv^\top) + \frac{-\gamma^2\cos\theta}{1 - \gamma^2\cos^2\theta}(vu^\top + uv^\top) \right)$$

$$= I - \frac{\gamma}{1 - \gamma^2 \cos^2\theta} \cdot (uu^\top + vv^\top) + \frac{\gamma^2\cos\theta}{1 - \gamma^2\cos^2\theta}(vu^\top + uv^\top).$$

First, observe that $(u + v)^\top \mathbf{J} uu^\top (u + v) = (1 + \cos\theta)^2$ for $\mathbf{J} \in \{uu^\top, vv^\top, vu^\top, uv^\top\}$. Therefore, $(u + v)^\top M(u + v)$ equals for $M := I - A$

$$2(1 + \cos\theta)^2 \cdot \frac{1}{1 - \gamma^2\cos^2\theta} \cdot (\gamma - \gamma^2\cos\theta) = \frac{\gamma(1 + \cos\theta)^2}{1 + \gamma\cos\theta}.$$

Let $M$ be $I - A$. Then

$$h = a^2 b^{-2} y^\top A y = a^2 b^{-2} (u + v)^\top (I - M)(u + v)$$

$$= a^2 b^{-2} \left( 2 + 2\cos\theta - \frac{2\gamma(1 + \cos\theta)^2}{1 + \gamma\cos\theta} \right)$$

$$= 2a^2 b^{-2}(1 + \cos\theta) \left( 1 - \frac{(1 + \cos\theta)\gamma}{1 + \gamma\cos\theta} \right)$$

$$= 2a^2 b^{-2}(1 + \cos\theta) \left( \frac{(1 - \gamma)}{1 + \gamma\cos\theta} \right) = 2a^2 b^{-1} \frac{(1 + \cos\theta)}{(1 + \gamma\cos\theta)}.$$

Overall, we get that $s_1 s_2 s_{1,2} = 1 - \gamma^2 \cos^2\theta$ and $h' - h = 2a^2 b^{-1} \frac{(1 - \gamma)\cos\theta}{1 + \gamma\cos\theta} = \frac{2a^2\cos\theta}{1 + \gamma\cos\theta}$.

$\square$

### A.3 Discrete Gaussian

**Fact A.5** (Translation of Discrete Gaussian). *For any $\theta \in \mathbb{R}$, $s > 0$, $\mu \in \mathbb{R}$, $\sigma \in \mathbb{R}_+$, the random variables $X \sim \mathsf{DG}[\mu, \sigma, \theta, s]$ and $X' := \sigma Y + \mu$ for $Y \sim \mathsf{DG}[0, 1, \theta', s']$ with $\theta' = (\theta - \mu)/\sigma$ and $s' = s/\sigma$ have the same law.*

*Proof.* First the support of both $X$ and $X'$ are equal to $\theta + s\mathbb{Z}$ (indeed the support of $Y$ is $\theta' + s'\mathbb{Z}$, which when multiplied by $\sigma$, yields $(\theta - \mu) + s\mathbb{Z}$, and further shifting by $\mu$ yields $\theta + s\mathbb{Z}$.

Starting with $X$, for any $i \in \mathbb{Z}$, $\mathbb{P}(X = \theta + si) \propto s\phi_{\mu,\sigma}(\theta + si) \propto \frac{s}{\sigma} \exp(-0.5(\theta + si - \mu)^2/\sigma^2) \propto \exp(-0.5(\theta + si - \mu)^2/\sigma^2)$, where we use that $s$ and $\sigma$ can be absorbed into the normalizing constant.

Turning to the random variable $X'$,

$$\mathbb{P}(X' = \theta + si) = \mathbb{P}(\sigma Y + \mu = \theta + si) = \mathbb{P}(Y = (\theta - \mu)/\sigma + si/\sigma) = \mathbb{P}(Y = \theta' + s'i)$$

$$\propto \exp(-0.5(\theta' + s'i)^2) \propto \exp\left( -0.5 \left( \frac{\theta + si - \mu}{\sigma} \right)^2 \right),$$

using the definitions of $\theta'$ and $s'$. Since the support is equal and the two distributions are equal up to constant, they must be equal.

$\square$

**Fact A.6** ([DKRS23, Fact C.3] and [DK22, Lemma 3.12]). We have the following:

- For any polynomial $p$ of degree at most $k$ and $\theta \in \mathbb{R}$ and $s > 0$, we have that
$$\left| \mathbb{E}_{G \sim \mathcal{N}(0,1)}[p(G)] - \mathbb{E}_{Y \sim \mathsf{DG}\left[0,1,\theta,s\right]}[p(Y)] \right| \lesssim \sqrt{\mathbb{E}_{G \sim \mathcal{N}(0,1)}[p^2(G)]} k! 2^{O(k)} \exp(-\Omega(1/s^2)).$$
- (Monomials and for the unnormalized measure) For any $k \in \mathbb{N}$ and $s \geq 0$:
$$\left| \mathbb{E}_{G \sim \mathcal{N}(0,1)}[G^k] - \mathbb{E}_{Y \sim \mathsf{DG}'\left[0,1,\theta,s\right]}[Y^k] \right| \lesssim k!(O(s))^k \exp(-\Omega(1/s^2)).$$ In particular, the total mass of $\mathsf{DG}'\left[0,1,\theta,s\right]$ is $1 \pm \exp(-\Omega(1/s^2))$.

# B  Proofs Deferred from Section 3

## B.1  Proof of Proposition 3.3

**Proposition 3.3.** *Testing Problem 3.1 is equivalent to Testing Problem 1.3 when $E = \mathsf{DG}\left[0,\sigma,0,s\right]$.*

*Proof.* First, by definition the distributions $P$ under Testing Problem 1.3 and Testing Problem 3.1 are the same. For $Q$, we shall do the calculations explicitly.

As a starting point, it is easy to see that the conditional distribution of $X$ given $y$ under Testing Problem 1.3 is an instance of NGCA, as in Testing Problem 3.1. To see this, define $x' = v^\top x$ to be the projection of $x$ along $v$, and define $x_\perp = x - (v^\top x)v$ to be its orthogonal projection. Observe that $x'$ and $x_\perp$ are distributed as standard (multivariate) Gaussian and are independent of each other (because $X \sim \mathcal{N}(0, \mathbf{I}_d)$). Hence, the conditional distribution of $y$ given $X \equiv (x', x_\perp)$ can be written as $y = \rho x' + Z$, implying that $y$ is independent of $x_\perp$. Therefore, the conditional distribution of $X$ given $y = y_0$ follows like a standard (multivariate) Gaussian in subspace orthogonal to $v$. Along the direction $v$, the distribution of $X$ is equivalent to the conditional distribution of $x'$ given $y$, which we denote by $\widetilde{J}_y$. Our goal is to show that $\widetilde{J}_y$ is equal to $\mathsf{DG}\left[\mu_y, \sigma, \theta_y, s'\right]$ as in Testing Problem 3.1.

Observe that marginal distribution of $Y$ is a Gaussian mixture with countable components, given by

$$Y \sim \sum_{i \in \mathbb{Z}} w(i) \mathcal{N}(si, \rho^2),$$

with $w(i) = cs(2\pi)^{-1/2} \exp(-s^2 i^2/(2\sigma^2))$, where $c$ denotes the normalization constant. Since $Z$ is discrete over the domain $s\mathbb{Z}$, the conditional distribution of $X$ given $Y = y_0$ is discrete with support $(y_0 - s\mathbb{Z})/\rho = \theta_{y_0} - s'\mathbb{Z}$, which is the same support as $\mathsf{DG}\left[\mu_{y_0}, \sigma, \theta_{y_0}, s'\right]$. For any $x_0$ in this discrete set, the conditional probability of $X = x_0$ given $y = y_0$ is given by the following (where we hide multiplicative terms that do not depend on $x_0$ under the normalization constant):

$$\begin{aligned}
\mathbb{P}(X = x_0 | X + Z = y_0) &\propto f_X(x_0) \mathbb{P}(Z = y_0 - \rho x_0) \\
&\propto \left( \exp(-x_0^2/2) \right) \left( w(y_0 - \rho x_0) \right) \\
&\propto \exp(-x_0^2/2) \exp\left( -(y_0 - \rho x_0)^2/(2\sigma^2) \right) \\
&\propto \exp\left( -\frac{1}{2} \left( x_0^2 + \frac{\rho^2 x_0^2}{\sigma^2} - \frac{2y_0 \rho x_0}{\sigma^2} \right) \right) \\
&\propto \exp\left( -\frac{1}{2} \left( \frac{x_0^2}{\sigma^2} - \frac{2y_0 \rho x_0}{\sigma^2} \right) \right) \\
&\propto \exp\left( -\frac{1}{2} \left( \frac{x_0}{\sigma} - \frac{y_0 \rho}{\sigma} \right)^2 \right) \\
&\propto \exp\left( -\frac{1}{2\sigma^2} (x_0 - y_0 \rho)^2 \right) \\
&\propto \exp\left( -\frac{1}{2\sigma^2} (x_0 - \mu_{y_0})^2 \right),
\end{aligned}$$

which is the mass assigned by $\text{DG}\big[\mu_{y_0}, \sigma, \theta_{y_0}, s'\big]$; Here we repeatedly use that $\rho^2 + \sigma^2 = 1$.

$\square$

## B.2 Concentration Properties of Distributions

In this section, we state the concentration properties of various distributions that appear in our analysis.

**Lemma B.1.** *Let the parameters be as in Definition 3.2. Then we have the following:*

1. *The distributions $A_y$, $\widetilde{A}_y$, and $B_y$ are $O(|y| + \sigma)$-subgaussian and if $X$ follows either one of these distributions, then $\mathbb{P}(|X - \mu_y| > t) \lesssim e^{-t^2/2\sigma^2}$.*

2. *The distribution $\text{DG}\big[0, \sigma, 0, s\big]$ is an $O(\sigma)$-subgaussian distribution, and $R_{\rho,E}$ is an $O(\sigma + \rho)$-subgaussian distribution.*

3. *The distributions $P_v^{A_y}$, $P_v^{\widetilde{A}_y}$, and $P_v^{B_y}$ are a $O(|y| + \sigma + 1)$-subgaussian distributions.[5]*

*Proof.* We do it case-by-case.

1. For $s'' := s'/\sigma$ and that $t \geq 1$:

$$
\begin{aligned}
\mathbb{P}_{X \sim A_y}&(|x - \mu_y| > t) \\
&= \mathbb{P}_{X \sim \text{DG}\big[0, 1, (\theta_y - \mu_y)/\sigma, s''\big]}(|W| > t/\sigma) && \text{(Fact A.5)} \\
&\lesssim \sum_{i \in \mathbb{N}} s'' \phi_{0,1}(\tfrac{t}{\sigma} + s'' i) + \sum_{i \in \mathbb{N}} s \phi_{0,1}(-\tfrac{t}{\sigma} - s'' i) && \text{(Fact 2.10 as } s'' \ll 1) \\
&\lesssim \sum_{i \in \mathbb{N}} s'' e^{-0.5\sigma^{-2}t^2 - 0.5 s''^2 i^2} && (\sigma \leq 1) \\
&\lesssim e^{-t^2/2\sigma^2} \sum_{i \in \mathbb{N}} s'' e^{-0.5 i^2 s''^2} \\
&\lesssim e^{-t^2/2\sigma^2} .
\end{aligned}
\tag{6}
$$

This tail also implies $O(|y| + \sigma)$-subgaussianity as follows: we claim that $P(|X| > t) \lesssim e^{-\frac{ct^2}{\max(|\mu_y|, \sigma)^2}}$. Observe that it suffices to consider $t \gtrsim \max(|\mu_y|, \sigma)$; otherwise, the bound is trivially true. For $t \gg |\mu_y|$, $\mathbb{P}(|X| > t) \leq \mathbb{P}(|X - \mu_y| \geq t/2)$ and we can then use (6). The same arguments hold for $B_y$. The claim for the tails of $|X|$ under $\widetilde{A}_y$ follows from that of $A_y$ because $\widetilde{A}_y$ is obtained from conditioning on an event of probability at least 0.5.

2. The claim for $\text{DG}\big[0, \sigma, 0, s\big]$ follows from (6). For $R_{\rho,E}$, we use the fact that if $x_1$ and $x_2$ are two independent $\sigma_1$ and $\sigma_2$-subgaussian random variables, then their sum is $O(\sqrt{\sigma_1 + \sigma_2})$-subgaussian [Ver18, Proposition 2.6.1].

3. After rotating appropriately, $P_v^{A_y}$ and $P_y^B$ are vectors of independent coordinates and thus follow a multivariate subgaussian distribution with variance proxy bounded by the subgaussian parameter of any individual coordinate [Ver18, Lemma 3.4.2]. The subgaussian proxy for the $v$ direction is established in the first item, while for the other coordinates it is $O(1)$.

$\square$

## B.3 Proof of Proposition 3.6

**Proposition 3.6** (SQ Hardness of Continuous Noise). *Consider Testing Problem 3.4. Then for any $m \in \mathbb{N}$ and $q \in \mathbb{N}$ satisfying $\frac{\rho^2 \sqrt{\log(1/q)}}{\sqrt{d}} \lesssim \frac{1}{m}$, we have that $\text{SDA}(m) \gtrsim q$.*

---

[5] A multivariate random vector $X$ is termed $\sigma$-subgaussian if, for all unit vectors $v$, the real-valued random variable $v^\top X$ is $\sigma$-subgaussian.

*Proof.* To calculate the average SQ correlation between $T_v$ and $T_{v'}$, we can first calculate the average correlation between the conditional distributions and then take the average marginal over $y$ to obtain the following expression:

$$\chi_{\mathcal{N}(0,\mathbf{I}_d)\times R}\left(T_v, T_{v'}\right) = \mathbb{E}_{y\sim R}\left[\chi_{\mathcal{N}(0,\mathbf{I}_d)}\left(P_v^{B_y}, P_{v'}^{B_y}\right)\right]. \tag{7}$$

Here, we crucially used that the marginal distribution of $y$ under $P$, $T_v$ and $T_{v'}$ is identical.

Observe that the distribution $P_v^{B_y}$ is equal to $\mathcal{N}(\mu_y v, (\mathbf{I}_d - vv^\top) + \sigma^2 vv^\top)$. Using Lemma A.4 with $a = \mu_y = \rho y$, $\gamma = 1 - \sigma^2 = \rho^2$, and $\cos\theta = v^\top v'$ to calculate $\chi_{\mathcal{N}(0,\mathbf{I}_d)}\left(P_v^{B_y}, P_{v'}^{B_y}\right)$, we obtain

$$1 + \chi_{\mathcal{N}(0,\mathbf{I}_d)}\left(P_v^{B_y}, P_{v'}^{B_y}\right) = \frac{\exp\left(\frac{\alpha^2\cos\theta}{1+\gamma\cos\theta}\right)}{\sqrt{1-\gamma^2\cos^2\theta}} = \frac{\exp\left(\frac{\rho^2 y^2\cos\theta}{1+\rho^2\cos\theta}\right)}{\sqrt{1-\gamma^2\cos^2\theta}}$$
$$= (1 + f(\theta))\exp\left(g(\theta)y^2\right)$$

for appropriately defined $f(\theta) := \frac{1}{\sqrt{1-\rho^4\cos^2\theta}} - 1$ and $g(\theta) := \frac{\rho^2\cos\theta}{1+\rho^2\cos\theta}$. Therefore, the average correlation over $y \in R$ is equal to

$$\chi_{\mathcal{N}(0,\mathbf{I}_d)\times R}\left(T_v, T_{v'}\right) = (1 + f(\theta))\,\mathbb{E}_{y\sim R}\left[\exp\left(g(\theta)y^2\right)\right] - 1. \tag{8}$$

Now, observe that $|g(\theta)| \leq \rho^2 \leq \rho_0^2$ by assumption for a sufficiently small $\rho_0$. Therefore, if we define $r(\theta) := \mathbb{E}_{y\sim R}\left[\exp\left(g(\theta)y^2\right)\right] - 1$, then Fact A.1 implies that

$$|r(\theta)| := \left|\mathbb{E}_{y\sim R}\left[\exp\left(g(\theta)y^2\right)\right] - 1\right| \lesssim |g(\theta)|.$$

Combining this with (8), we obtain

$$\left|\chi_{\mathcal{N}(0,\mathbf{I}_d)\times R}\left(T_v, T_{v'}\right)\right| = (1 + f(\theta))(1 + r(\theta)) - 1 \lesssim f(\theta) + |r(\theta)| \qquad \text{(using } |r(\theta)| \lesssim 1\text{)}$$
$$\lesssim \rho^4\cos^2\theta + \rho^2|\cos\theta| \lesssim \rho^2|\cos\theta|, \tag{9}$$

where we use that $\rho^2|\cos\theta| \leq 0.1$. In particular,

$$\chi^2\left(T_v, \mathcal{N}(0,\mathbf{I}_d)\times R\right) \lesssim \rho^2. \tag{10}$$

We are now ready to show that $\mathrm{SDA}(m) \geq q$, for which we need to show the following:

$$\sup_{\mathcal{E}:\mathbb{P}_{v,v'}((v,v')\in\mathcal{E})\geq 1/q^2} \mathbb{E}_{v,v'}\left[\left|\chi_P\left(T_v, T_{v'}\right)\right|\big|\mathcal{E}\right] \leq \frac{1}{m}.$$

Using (9), it suffices to show that

$$\sup_{\mathcal{E}:\mathbb{P}_{v,v'}((v,v')\in\mathcal{E})\geq 1/q^2} \mathbb{E}_{v,v'}\left[\left|\rho^2|v^\top v'|\right|\big|\mathcal{E}\right] \leq \frac{1}{m}. \tag{11}$$

If $v$ and $v'$ are two independent random unit vectors, then $W := v^\top v'$ is a centered $\Theta(1/\sqrt{d})$-subgaussian random variable [Ver18, Theorem 3.4.6]. For subgaussian random variables, we use the simple inequality (a simple consequence of Hölder's inequality) $\mathbb{E}[|W|\big|\mathcal{E}] \leq \|W\|_{\psi_2}\sqrt{\log(1/\mathbb{P}(\mathcal{E}))} \lesssim \frac{1}{\sqrt{d}}\sqrt{\log(q)}$. We obtain that the left hand side in (11) is less than $\rho^2\left(\frac{O(1)}{\sqrt{d}}\cdot\sqrt{\log q}\right)$ and hence (11) holds if

$$\frac{\rho^2\sqrt{\log(1/q)}}{\sqrt{d}} \lesssim \frac{1}{m}, \tag{12}$$

which is the desired conclusion.

$\square$

### B.4 Proof of Proposition 3.9

Observe that Proposition 3.9 follows from the result below because of Lemma B.1. Indeed, Lemma B.1 implies that (i) $\mathbb{P}_y(|y| \geq L) \lesssim e^{-\Omega(L^2)}$ as $\sigma \lesssim 1$ and $\rho \lesssim 1$ and (ii) for any $y$ with $|y| \leq d/2$, $\mathbb{P}_{A_y}(|z| > d) \leq \mathbb{P}_{A_y}(|z - \mu_y| > d/2) \lesssim e^{-\Omega(d^2)}$.

**Proposition B.2.** *Let $f : \mathcal{U} \to [0,1]$. For $L \geq 1$, define the set $\mathcal{C} : \{y : |y| \leq L\widetilde{\sigma}\}$ and the function $\widetilde{f} := f\mathbb{1}_{y \in \mathcal{C}}$. Then for any $\ell \in \mathbb{N}$, we have that for $y \sim R_{\rho,E}$:*

$$|\mathbb{E}_{T_v}[f] - \mathbb{E}_{Q_v}[f]| \leq 4\mathbb{P}(y \notin \mathcal{C}) + \max_{y : |y| \leq L} \mathbb{P}_{A_y}(|z| > d) + \Big| \mathbb{E}_y \Big[ \sum_{k=1}^{\ell} \Big( \widetilde{\mathbf{A}}_{k,y} - \mathbf{B}_{k,y} \Big) \langle v^{\otimes k}, \mathbf{T}_{k,y} \rangle \Big] \Big|$$

$$+ \Big| \mathbb{E}_{\widetilde{Q}_v}[\widetilde{f}^{>\ell}] - \mathbb{E}_{Q_v}[\widetilde{f}^{>\ell}] \Big|, \tag{13}$$

*where $\mathbf{T}_{k,y} := \mathbb{E}_{x \sim \mathcal{N}(0, \mathbf{I}_d)}[\widetilde{f}_y(x) H_k(x)]$, $\mathbf{A}_{k,y} := \mathbb{E}_{x \sim \widetilde{A}_y}[h_k(x)]$ and $\mathbf{B}_{k,y} := \mathbb{E}_{x \sim B_y}[\widetilde{f}(x)]$ for $A_y, B_y$ defined in Testing Problems 3.1 and 3.4.*

*Proof.* We start by replacing $\mathbb{E}_{Q_v}[f]$ with $\mathbb{E}_{\widetilde{Q}_v}[f]$ at the cost of additive $\mathrm{TV}(Q_v, \widetilde{Q}_v)$. This total variation distance is $O(\mathbb{P}_{Q_v}(|v^\top x| > d))$, which can be upper bounded by $\mathbb{P}(|y| \geq L) + \max_{y : |y| \leq L} \mathbb{P}(|v^\top x| \geq d)$. Hence, in the rest of this proof, we shall use $\widetilde{Q}$ everywhere.

Next we decompose $f = \widetilde{f} + f'$ for $f' := f\mathbb{1}_{y \notin \mathcal{C}}$.

Then we further decompose $\widetilde{f}$ as $\widetilde{f}^{\leq \ell} + \widetilde{f}_y^{>\ell}$. By triangle inequality, it suffices to show that the expectations of $\widetilde{f}^{\leq \ell}, \widetilde{f}^{>\ell)}$, and $f'$ are close. Observe that the term for $\widetilde{f}^{>\ell}$ is already present in the final conclusion. Next, for $f'$, the boundedness of $f$ and the same marginals of $Q_v$ and $T_v$ imply that

$$\Big| \mathbb{E}_{\widetilde{Q}_v}[f'] - \mathbb{E}_{T_v}[f'] \Big| \leq 2\mathbb{P}(y \notin \mathcal{C}).$$

In the remainder, we focus on the terms corresponding to $\widetilde{f}^{\leq \ell}$. By the law of total expectation (whose validity for $\widetilde{f}^{\leq \ell}$ is justified below), we have that

$$\mathbb{E}_{\widetilde{Q}_v}[\widetilde{f}_y^{\leq \ell}(x, y)] = \mathbb{E}_y \Big[ \mathbb{E}_{x \sim P_v^{A_y}} \Big[ \widetilde{f}_y^{\leq \ell} \Big] \Big]. \tag{14}$$

To compute the inner expectation, which is an instance of the unsupervised NGCA, we will use [DKRS23, Lemma 3.3]:

**Lemma B.3** (Fourier Decomposition Lemma of [DKRS23]). *Let $A'$ be any distribution supported on $\mathbb{R}$ and $v$ a unit vector. Then for any $\ell \in \mathbb{N}$ and $g : \mathbb{R}^d \to [0,1]$,*

$$\mathbb{E}_{x \sim P_v^{A'}}[g^{\leq \ell}(x)] = \sum_{k=0}^{\ell} \mathbf{A}_k \langle v^{\otimes k}, \mathbf{T}_k \rangle,$$

*where $\mathbf{A}_k = \mathbb{E}_{x \sim A'}[h_k(x)]$ and $\mathbf{T}_k = \mathbb{E}_{x \sim \mathcal{N}(0, \mathbf{I}_d)}[g(x) \mathbf{H}_k(x)]$.*

Consider a fixed $y_0 \in \mathbb{R}$ and apply the above result to $A' := \widetilde{A}_{y_0}$ and $g(x) := \widetilde{f}_y(x) := f(x, y_0)\mathbb{1}_{y_0 \in \mathcal{C}}$. Define $\widetilde{\mathbf{A}}_{k,y} := \mathbb{E}_{x \sim \widetilde{A}_y}[h_k(x)]$ and $\mathbf{B}_{k,y} := \mathbb{E}_{x \sim B_y}[h_k(x)]$ and $\mathbf{T}_{k,y} := \mathbb{E}_{x \sim \mathcal{N}(0, \mathbf{I}_d)}[\widetilde{f}(x, y) \mathbf{H}_k(x)]$. We obtain that

$$\mathbb{E}_{\widetilde{Q}_v}[\widetilde{f}^{\leq \mathcal{L}}] = \mathbb{E}_y \Big[ \mathbb{E}_{x \sim P_v^{\widetilde{A}_y}} \Big[ \widetilde{f}_y^{\leq \ell}(x) \Big] \Big] = \mathbb{E}_y \Big[ \sum_{k=0}^{\ell} \widetilde{\mathbf{A}}_{k,y} \langle v^{\otimes k}, \mathbf{T}_{k,y} \rangle \Big].$$

Observe that the term $k = 0$ corresponds to $\mathbb{E}_{\mathcal{N}(0, \mathbf{I}_d)}[f(x, y_0)]$ for each $y_0$, implying that the expectation of the $k = 0$ term (over $y$) is exactly $\mathbb{E}_P[\widetilde{f}(x, y)]$. Thus, we get the following decomposition:

$$\mathbb{E}_{\widetilde{Q}_v}[\widetilde{f}^{\leq \mathcal{L}}] - \mathbb{E}_P[\widetilde{f}] = \mathbb{E}_{y \sim R'} \Big[ \sum_{k=1}^{\ell} \widetilde{\mathbf{A}}_{k,y} \langle v^{\otimes k}, \mathbf{T}_{k,y} \rangle \Big]. \tag{15}$$

Similarly, the decomposition for the continuous Gaussian noise is as follows:

$$\mathbb{E}_{T_v}[\widetilde{f}_y^{\leq \ell}(x)] - \mathbb{E}_P[\widetilde{f}] = \mathbb{E}_{y \sim R'}\left[\sum_{k=1}^{\ell} \mathbf{B}_{k,y}\left\langle v^{\otimes k}, \mathbf{T}_{k,y}\right\rangle\right]. \tag{16}$$

The claim follows by combining Equations (15) and (16).

**Justifying (14).** It suffices to show that $\mathbb{E}_{\widetilde{Q}_v}[|\widetilde{f}^{\leq \ell}|] < \infty$, which we will establish below. By Fubini's theorem, we have that

$$\mathbb{E}_{\widetilde{Q}_v}[|\widetilde{f}^{\leq \ell}|] = \mathbb{E}_y\left[\mathbb{E}_{x \sim P_v^{\widetilde{A}_y}}\left[\left|\widetilde{f}_y^{\leq \ell}\right|\right]\right]$$

$$\leq \mathbb{E}_y\left[\mathbb{E}_{x \sim P_v^{\widetilde{A}_y}}\left[\left|\sum_{k=0}^{\ell}\langle[\mathbb{E}_{x' \sim P_v^{\widetilde{A}_y}} f(x')\mathbf{H}_k(x')], \mathbf{H}_k(x)\rangle\right|\right]\right]$$

$$\leq \sum_{k=0}^{\ell}\mathbb{E}_y\left[\mathbb{E}_{x \sim P_v^{\widetilde{A}_y}}\left[\left|\langle[\mathbb{E}_{x' \sim P_v^{\widetilde{A}_y}} f(x')\mathbf{H}_k(x')], \mathbf{H}_k(x)\rangle\right|\right]\right].$$

This can be further upper bounded by finite sum of the terms (at most $d^{O(\ell)}$) involving

$$\mathbb{E}_y\mathbb{E}_{x \sim P_v^{\widetilde{A}_y}}[|\mathbb{E}_{x'} f(x')p(x')| \cdot |p(x)|]$$

for some polynomials $p(\cdot)$. Since $|f|$ is upper bounded by 1, the term above is further upper bounded by $\mathbb{E}_y\mathbb{E}_{x \sim P_v^{\widetilde{A}_y}}[|p(x)|^2]$ using Jensen's inequality. Using Lemma B.1, $\mathbb{E}_{x \sim P_v^{\widetilde{A}_y}}[|p(x)|^2]$ is upper bounded by $\mathrm{poly}(|\mu_y|, d, \|p\|_{\ell_2})$ and since $\mu_y$ is linear in $y$, $\mathbb{E}[\mathrm{poly}(\mu_y)]$ is also finite because $R$ is $O(1)$-subgaussian.

A similar reasoning justifies (14) for $T_v$. $\qquad\square$

## B.5   Proof of Lemma 3.10

**Lemma 3.10** (Closeness of Hermite Coefficients). *For any $y \in \mathbb{R}$ and $k \in \mathbb{N}$, we have*
- *(Tighter for small k)* $|\widetilde{\mathbf{A}}_{k,y} - \mathbf{B}_{k,y}| \lesssim \max\left(1, |\mu_y|^k\right) k^{O(k)} \cdot \left(e^{-\Omega\left(\frac{\rho^2}{s^2}\right)} + e^{-\Omega(d)}\right)$.
- *(Tighter for larger k)* $|\widetilde{\mathbf{A}}_{k,y} - \mathbf{B}_{k,y}| \lesssim e^{O(\mu_y^2)}$.

*Proof.* We first consider the case for large $k$.

**Large $k$.** For large $k$, we shall use the fact that $|h_k(x)| \leq \exp(x^2/4)$ for all $x \in \mathbb{R}$ [Kra04]. Lemma B.1 implies that for both $B_y$ and $\widetilde{A}_y$,

$$\forall t: \qquad P(|x - \mu_y| \geq t) \leq O(1)\exp(-x^2/2),$$

where we use that $\sigma \leq 1$. Therefore, under the both $X \sim \widetilde{A}_y$ and $X \sim B_y$, we have that

$$\mathbb{P}(|X| > t) \leq O(1)\exp(O(\mu_y^2))\exp(-0.4t^2). \tag{17}$$

Indeed for $t \leq 10\mu_y$, the upper bound is bigger than 1 and hence holds; for $t \geq 10\mu_y$, $\mathbb{P}(|X| > t) \leq \mathbb{P}(|X - \mu_y| \geq 0.9t) \lesssim \exp(-0.4t^2)$.

Therefore, we can upper bound $\mathbb{E}[|h_k(X)|]$ for both distributions as follows:

$$\mathbb{E}[|h_k(X)|] \leq \mathbb{E}[e^{X^2/4}] \leq 1 + \int_1^\infty \mathbb{P}(|X| > 2\sqrt{\log_e u})du \lesssim 1 + \int_1^\infty e^{-0.4 \cdot 4 \cdot \log_e u}$$

$$\lesssim \exp(O(\mu_y^2))\left(1 + \int_1^\infty u^{-1.6}du\right) \lesssim \exp(O(\mu_y^2)).$$

**Smaller $k$.** We first define $\mathbf{C}_{k,y} := \mathbb{E}_{x \sim A_y}[h_k(x)]$. Since $\widetilde{A}_y$ is $A_y$ conditioned on $\mathcal{E} := \{z : |z| \leq d\}$ and satisfies $\mathbb{P}(\mathcal{E}) \geq 1 - \tau$ for $\tau \lesssim e^{-\Omega(d)}$ (see Lemma B.1), we have that for any function $g$:

$$\left| \mathbb{E}_{\widetilde{A}_y}[g] - \mathbb{E}_{B_y}[g] \right| \lesssim 2 \left| \mathbb{E}_{A_y}[g] - \mathbb{E}_{B_y}[g] \right| + \tau \mathbb{E}_{B_y}[g] + \sqrt{\tau \mathbb{E}_{A_y}[g^2]} .$$

The above inequality follows by noting that the left hand side above is exactly equal to $\frac{\mathbb{E}_{A_y}[g] - \mathbb{E}_{B_y}[g]}{1-\tau} + \frac{\tau \mathbb{E}_{B_y}[g]}{1-\tau} + \frac{\mathbb{E}_{A_y}[g \mathbb{I}_{\mathcal{E}}]}{1-\tau}$ and then applying Cauchy-Schwarz inequality. In our context, the above display equation yields:

$$|\mathbf{B}_{k,y} - \mathbf{A}_{k,y}| \leq 2 |\mathbf{B}_{k,y} - \mathbf{C}_{k,y}| + \tau |\mathbf{B}_{k,y}| + \sqrt{\tau} \sqrt{\mathbb{E}_{A_y}[h_k^2(x)]}. \tag{18}$$

We will now upper bound this difference. We first claim that for $\widetilde{\theta}_y = (\theta_y - \mu_y)/\sigma$ and $\widetilde{s} = s'/\sigma$, we have that

$$\mathbf{B}_{k,y} - \mathbf{C}_{k,y} = \mathbb{E}_{x \sim \mathcal{N}(0,1)}[h_k(\sigma x + \mu_y)] - \mathbb{E}_{x' \sim \mathsf{DG}\left[0, 1, \widetilde{\theta}_y, \widetilde{s}\right]}[h_k(\sigma x' + \mu_y)]. \tag{19}$$

To see this, recall that $\mathbf{B}_{k,y} = \mathbb{E}_{x \sim B_y}[h_k(x)] = \mathbb{E}_{x \sim \mathcal{N}(\mu_y, \sigma^2)} h_k(x)$, which implies that it is equal to $\mathbb{E}_{x \sim \mathcal{N}(0,1)} h_k(\sigma x + \mu_y)$. For $\mathbf{A}_{k,y}$, the claim follows analogously from Fact A.5.

**Lemma B.4.** *Let $k \in \mathbb{N}$, $q \in \mathbb{R}$, $a \in \mathbb{R}$, $b \in \mathbb{R}$ and $s'' \ll 1$. Let $G \sim \mathcal{N}(0,1)$ and $Y \sim \mathsf{DG}\left[0, 1, q, s'\right]$.*

- $\left| \mathbb{E}[h_k(b + aG)] - \mathbb{E}[h_k(b + aY)] \right| \leq \max(1, |b|^k) \max(1, |a|^k) k^{O(k)} e^{-\frac{1}{s''^2}}$ .

- $|\mathbb{E}[|h_k(b + aG)|]|^2 \leq \mathbb{E}[|h_k(b + aG)|^2] \leq k^{O(k)} \max(1, b^{2k}) \max(1, a^{2k})$.

- $\mathbb{E}[|h_k(b + aY)|^2] \leq k^{O(k)} \max(1, b^{2k}) \max(1, a^{2k})$.

Applying this result on (19) with $b = \mu_y$, $a = \sigma \leq 1$ and $s'' = \widetilde{s} = s'/\sigma = s/\rho\sigma$ and plugging it in (18) in combination with $\tau \lesssim e^{-\Omega(d)}$, we get Lemma 3.10.

$\square$

We now provide the proof of Lemma B.4

*Proof.* Defining the polynomial $p_k(x) := h_k(b + ax)$, we can apply Fact 2.10 to $p_k(\cdot)$ to conclude that the deviation in the first item is at most

$$\sqrt{\mathbb{E}_{G \sim \mathcal{N}(0,1)}[h_k^2(b + aG)]} k! 2^{O(k)} \exp(-\Omega(1/s^2)).$$

Hence, to establish both the first and the second items, it remains to show the upper bound $\sqrt{\mathbb{E}_{G \sim \mathcal{N}(0,1)}[h_k^2(b + aG)]} \lesssim k^{O(k)} \max(1, |b|^k) \max(1, |a|^k)$. To that effect, we use the explicit form of the Hermite polynomials:

$$h_k(x) := \sqrt{k!} \sum_{\ell=0}^{\lfloor k/2 \rfloor} \frac{(-1)^\ell}{\ell!(k - 2\ell)!} \frac{1}{2^\ell} x^{k-2\ell},$$

which gives the following expression:

$$\mathbb{E}[h_k^2(b + aG)] = k! \mathbb{E}\left[ \sum_{\ell=0, \ell'=0}^{\lfloor k/2 \rfloor} \frac{(-1)^\ell}{\ell!(k - 2\ell)!} \frac{1}{2^\ell} (b + aG)^{k-2\ell} \frac{(-1)^{\ell'}}{\ell'!(k - 2\ell')!} \frac{1}{2^{\ell'}} (b + aG)^{k-2\ell'} \right].$$

There are $\Theta(k^2)$ terms in the expression above and by linearity of the expectation, it suffices to control the maximum term above:

$$\mathbb{E}[h_k^2(b + aG)] \leq k^2 k! \max_{\ell \leq k/2, \ell' \leq k/2} \mathbb{E}\left[ (b + aG)^{k-2\ell}(b + aG)^{k-2\ell'} \right] \tag{20}$$

$$\leq k^2 k! \max_{\ell \leq k, \ell' \leq k} \sqrt{\mathbb{E}[(b+aG)^{2\ell}]} \sqrt{\mathbb{E}[(b+aG)^{2\ell'}]}$$

$$\leq k^2 k! \max_{\ell \leq k} \mathbb{E}[(b+aG)^{2\ell}]$$

$$\leq k^2 k! \max_{\ell \leq k} \mathbb{E}[2^{2\ell} b^{2\ell} + 2^{2\ell} a^{2\ell} G^{2\ell}]$$

$$\leq 2^{2k} k^2 k! \max_{\ell \leq k} \mathbb{E}[b^{2\ell} + a^{2\ell} G^{2\ell}]$$

$$\leq 2^{2k} k^2 k! \max_{\ell \leq k} \mathbb{E}[b^{2\ell} + (O(\sqrt{k}))^k a^{2\ell}]$$

$$\leq k^{O(k)} \max(1, b^{2k}) \max(1, a^{2k}),$$

which proves the desired result.

We now focus on the third item. Here, we again apply Fact 2.10 but this time to the polynomial $p_k^2$, which would then imply that

$$\mathbb{E}[|h_k(b+aY)|^2] \leq \mathbb{E}[|h_k(b+aG)^2] + \sqrt{\mathbb{E}[|h_k(b+aG)^4]}(2k)^{O(k)} \exp(-\Omega(1/s^2))$$

$$\leq k^{O(k)} \left( \max(1, b^{2k}) \max(1, a^{2k}) + \sqrt{\mathbb{E}[|h_k(b+aG)^4]} \right).$$

To upper bound $\mathbb{E}[|h_k(b+aG)^4]$, we can use a similar series of arguments as in (21) to get the desired result, wherein we replace the use of Cauchy-Schwarz inequality with the inequality $\mathbb{E}[X_1 X_2 X_3 X_4] \leq \prod_{i=1}^{4} (\mathbb{E}[X_i^4])^{1/4}$. $\qquad\square$

We now provide the proof of Lemma B.4

*Proof.* Defining the polynomial $p_k(x) := h_k(b+ax)$, we can apply Fact 2.10 to $p_k(\cdot)$ to conclude that the deviation in the first item is at most

$$\sqrt{\mathbb{E}_{G \sim \mathcal{N}(0,1)}[h_k^2(b+aG)]} k! 2^{O(k)} \exp(-\Omega(1/s^2)).$$

Hence, to establish both the first and the second items, it remains to show the upper bound $\sqrt{\mathbb{E}_{G \sim \mathcal{N}(0,1)}[h_k^2(b+aG)]} \lesssim k^{O(k)} \max(1, |b|^k) \max(1, |a|^k)$. To that effect, we use the explicit form of the Hermite polynomials:

$$h_k(x) := \sqrt{k!} \sum_{\ell=0}^{\lfloor k/2 \rfloor} \frac{(-1)^\ell}{\ell!(k-2\ell)!} \frac{1}{2^\ell} x^{k-2\ell},$$

which gives the following expression:

$$\mathbb{E}[h_k^2(b+aG)] = k! \mathbb{E}\left[ \sum_{\ell=0, \ell'=0}^{\lfloor k/2 \rfloor} \frac{(-1)^\ell}{\ell!(k-2\ell)!} \frac{1}{2^\ell} (b+aG)^{k-2\ell} \frac{(-1)^{\ell'}}{\ell'!(k-2\ell')!} \frac{1}{2^{\ell'}} (b+aG)^{k-2\ell'} \right].$$

There are $\Theta(k^2)$ terms in the expression above. Moreover, By linearity of the expectation, it suffices to control the maximum term above:

$$\mathbb{E}[h_k^2(b+aG)] \leq k^2 k! \max_{\ell \leq k/2, \ell' \leq k/2} \mathbb{E}\left[ (b+aG)^{k-2\ell} (b+aG)^{k-2\ell'} \right] \tag{21}$$

$$\leq k^2 k! \max_{\ell \leq k, \ell' \leq k} \sqrt{\mathbb{E}[(b+aG)^{2\ell}]} \sqrt{\mathbb{E}[(b+aG)^{2\ell'}]}$$

$$\leq k^2 k! \max_{\ell \leq k} \mathbb{E}[(b+aG)^{2\ell}]$$

$$\leq k^2 k! \max_{\ell \leq k} \mathbb{E}[2^{2\ell} b^{2\ell} + 2^{2\ell} a^{2\ell} G^{2\ell}]$$

$$\leq 2^{2k} k^2 k! \max_{\ell \leq k} \mathbb{E}[b^{2\ell} + a^{2\ell} G^{2\ell}]$$

$$\leq 2^{2k} k^2 k! \max_{\ell \leq k} \mathbb{E}[b^{2\ell} + (O(\sqrt{k}))^k a^{2\ell}]$$

$$\leq k^{O(k)} \max(1, b^{2k}) \max(1, a^{2k}),$$

which proves the desired result.

We now focus on the third item. Here, we again apply Fact 2.10 but this time to the polynomial $p_k^2$, which would then imply that

$$\mathbb{E}[|h_k(b+aY)|^2] \leq \mathbb{E}[|h_k(b+aG)^2] + \sqrt{\mathbb{E}[|h_k(b+aG)^4]}(2k)^{O(k)} \exp(-\Omega(1/s^2))$$
$$\leq k^{O(k)} \left( \max(1, b^{2k}) \max(1, a^{2k}) + \sqrt{\mathbb{E}[|h_k(b+aG)^4]} \right).$$

To upper bound $\mathbb{E}[|h_k(b+aG)^4]$, we can use a similar series of arguments as in (21) with the inequality $\mathbb{E}[X_1 X_2 X_3 X_4] \leq \prod_{i=1}^{4}(\mathbb{E}[X_i^4])^{1/4}$ instead of Cauchy-Schwarz inequality to get the desired result. $\qquad\square$

## B.6 Proof of Proposition 3.11

**Proposition 3.11.** *Let* $\{\mathbf{T}_{k,y}\}_{k \in \mathbb{N}, y \in \mathbb{R}}$ *be tensors with* $\|\mathbf{T}_{k,y}\|_2 \leq 1$ *for all* $k \in \mathbb{N}, y \in \mathbb{R}$, *and let* $t \in \mathbb{N}$ *be arbitrary. Then for any* $\delta \in (0,1)$, *it holds with probability* $1-\delta$ *over a random unit vector* $v$ *that*

$$\mathbb{E}_y\big[\sum_{k=1}^{t} |\langle v^{\otimes k}, \mathbf{T}_{k,y}\rangle|\big] \lesssim t \quad and \quad \sum_{k>t+1}^{\infty} \mathbb{E}_y\left[|\langle v^{\otimes k}, \mathbf{T}_k\rangle|\right] \lesssim d^{O(1)}\Big(\frac{t\log\frac{t}{\delta}}{d}\Big)^{t/4} + d^{O(1)} \cdot \frac{1}{\delta}e^{-\frac{Cd}{\log\frac{d}{\delta}}}.$$

The result for $k \leq t$ follows by the claim that $\mathbf{T}_{k,y}$ has norm at most 1 almost surely. Hence, we will focus on the second claim, for which we shall crucially use the concentration results from [DKRS23], which we state in a different formulation below.

**Lemma B.5** (Lemma 3.7 and Corollary 3.9 in [DKRS23]). *Let* $\mathbf{T}_{k,y}$ *be a random* $k$-*tensor supported with randomness* $y$. *For a random unit vector* $v$ *independent of* $y$, *let* $W_k$ *denote the random variable* $\mathbb{E}_y|\langle v^{\otimes k}, \mathbf{T}_{k,y}\rangle|$. *Let* $W'$ *be the random variable* $v^\top w$ *for a unit vector* $w \in \mathcal{S}^{d-1}$. *Then for any even* $p \in \mathbb{N}$,

$$\|W_k\|_{L_p} \leq (\mathbb{E}_y\|\mathbf{T}_{k,y}\|_2^p)^{1/p}\|W'\|_{L_{pk/2}}^{k/2}. \tag{22}$$

*In particular, if* $\langle \mathbf{T}_{k,y}, \mathbf{T}_{k,y}\rangle \leq 1$ *almost surely, then there exists a constant* $C > 0$ *such that the following conclusion holds for any even* $p \in \mathbb{N}$ *and* $k \in \mathbb{N}$:

1.  $\|W_k\|_{L_p} \leq \left(\frac{Cpk}{d}\right)^{k/4}.$            *(useful for moderate* $k$: $k = o(d)$)

2.  $\|W_k\|_{L_p} \lesssim \min(d,k)^{1/p} \exp\left(-C\frac{kd}{\max(d,pk)}\right).$    *(useful for large* $k$: $k \asymp d$)

3.  $\|W_k\|_{L_p} \leq C\left(\frac{d}{pk}\right)^{d/p}.$                *(useful for extremely large* $k$: $k = \omega(d)$)

While (22) is established in [DKRS23, Lemma 3.7] for a fixed tensor $\mathbf{T}$, the desired follows by Jensen's inequality: for any even $p$ and error bound $g(v, y)$, we have that $\mathbb{E}_v(\mathbb{E}_y[g(v, y)])^p \leq \mathbb{E}_v\mathbb{E}_y[g(v,y)^p] = \mathbb{E}_y(\mathbb{E}_v[g(v,y)^p])$, where one can now use [DKRS23, Lemma 3.7].

We now couple Lemma B.5 with the simple fact that for any random variable $X$, with probability $1 - \delta$, $|X| \leq (1/\delta)^{1/p}\|X\|_{L_p}$ for any $p \geq 1$. Therefore, for any $k \geq t$, with probability $1 - \delta/k^2$, $|W_k|$ is less than $c\min(\|W_k\|_{L_{\log(k/\delta)}}, \sqrt{k/\delta}\|W_k\|_{L_2})$. We now calculate these bounds separately for different $k$.

- (Small $k$ and large $p$) Define $p_k \asymp \log(k/\delta)$. For any $k$ such that $k \geq t$ and $kp_k \leq C'dp_k$, we have

$$\|W_k\|_{L_{p_k}} \leq \left(\frac{Ct\log(t/\delta)}{d}\right)^{t/4} + \exp\left(-C\frac{d}{\log^2(d/\delta)}\right).$$

This follows by considering the following two regimes separately:

- $(c'p_k k \leq d$ for a tiny enough constant $c'$) In this regime, the bound $\left(\frac{Ckp_k}{d}\right)^{k/4}$ is decreasing in $k$ and thus the maximum is achieved at $k = t$.
- $(C'dp_k \geq c'p_k k \geq d$ for a large constant $c'$) In this regime, the second bound gives the desired result by noting $\max(d, p_k k) \leq C'd_p k$ and $k \geq d/p_k \geq d/p_d$.

- (Large $k$ and $p = 2$) Moreover, if $k \geq C'd$, then $\|W_k\|_2 \leq (\frac{d}{2k})^{d/2}$. Therefore, with probability $1 - \delta/k^2$, $|W_k| \leq \sqrt{(k^2/\delta)}(d/2k)^{d/2} \leq \sqrt{d/\delta}(d/2k)^{d/4}$.

Taking a union bound, we obtain the following bound that holds with probability at least $1 - \delta$,

$$\sum_{k>t} |W_k| = \sum_{k \in [t, C'd]} k^a |W_k| + \sum_{k \in [t, C'p_k]} k^a |W_k|$$

$$\lesssim (C'd)^1 \cdot \left( \left( \frac{t \log (t/\delta)}{d} \right)^{t/4} + d \exp\left( -C \frac{d}{\log(d/\delta)} \right) \right) + \sum_{k \geq C'd} \sqrt{d/\delta} \left( \frac{d}{2k} \right)^{d/2}.$$

The summation $\sum_{k \geq C'd} \left( \frac{d}{2k} \right)^{d/2}$ can be upper bounded by a constant factor multiple of the first expression in the sum (this can be seen by integrating $\int_{x \geq x_0} x^{-a} dx$ for $a > 2$), and the first expression is at most $e^{-\Omega(d)}$ because $C'$ is large enough.

## B.7 Proof of Theorem 3.7

We are now ready to present the proof of Theorem 3.7.

*Proof of Theorem 3.7.* Combining Proposition 3.9 with Lemma 3.10 and Proposition 3.11 and the fact that $|\mu_y| \leq L$ for $L \geq 1$, we obtain that for any $t \in \mathbb{N}$ and $\ell \in \mathbb{N}$ with probability at least $1 - d^{-\log^2 d}$,

$$\left| \mathbb{E}_{T_v}[f] - \mathbb{E}_{Q_v}[f] \right| \lesssim e^{-\Omega(L^2)} + e^{-\Omega(d)} + L^t t^{O(t)} \left( e^{-\Omega(\rho^2/s^2)} + e^{-\Omega(d)} \right) t$$

$$+ e^{cL^2}(dt)^{O(1)} \left( \frac{t \log t \log^3 d}{d} \right)^{t/4} + e^{cL^2} e^{-d/\text{polylog}(d)} + \left| \mathbb{E}_{T_v}[\widetilde{f}^{>\ell}] - \mathbb{E}_{\widetilde{Q}_v}[\widetilde{f}^{>\ell}] \right|.$$

For $L = \log^5 d$, $t = L^6$ and $\rho = st^2$, the sum of all but the last term is at most $O(e^{-L^2}) \leq d^{-\log^2(d/\alpha)}$. For the last term, we show in Appendix B.8 that taking $\ell$ large enough suffices—this argument uses Fact 2.1 and the truncation of $A_y$ as per [DKRS23]. $\square$

## B.8 Handling $\widetilde{f}^{>\ell}$

We now show that for any $f : \mathbb{R}^d \times \mathbb{R} \to [0, 1]$ and any $\delta \in (0, 1)$, there exists $\ell \in \mathbb{N}$, depending only on $(f, d, \delta, L, \sigma, \rho, s, \alpha)$ such that with $1 - \delta$, $|\mathbb{E}_{Q_v}[\widetilde{f}^{>\ell}] - \mathbb{E}_{T_v}[\widetilde{f}^{>\ell}]|$ is smaller than $\gamma$ for a $\gamma$ appropriately small.

First by Fact 2.1, we know that there exists an $\ell(\gamma')$ so that $\|\widetilde{f}^{>\ell}\|_{L_2(P)} \leq \gamma'$. Since $\chi^2(P, T_v)$ is finite (as established in (10)), this implies that $\|\widetilde{f}^{>\ell}\|_{L_2(T_v)}$ is also sufficiently small. By Cauchy-Schwarz, we get that for every $\gamma$ and $v \in \mathcal{S}^{d-1}$, there exists an $\ell'(\delta, d, \gamma)$ so that $|\mathbb{E}_{T_v}[\widetilde{f}^{>\ell}]| \leq \gamma$.

Thus, it remains to argue about $\mathbb{E}_{Q_v}[\widetilde{f}^{>\ell}]$. By a Markov inequality, it suffices to show that $\mathbb{E}_v[|\mathbb{E}_{Q_v}[\widetilde{f}^{>\ell}]|] \leq \mathbb{E}_v \mathbb{E}_{Q_v}[|\widetilde{f}^{>\ell}|] \leq \gamma/\delta$. Let $D$ be the distribution of $(x, y)$ obtained over $(x, y)$ as follows: first $y \sim R$ and then $v \sim \mathcal{S}^{d-1}$ and $x \sim P_v^{\widetilde{A}_y}$. Let $D_y$ be the conditional distribution of $x$ given $y$ under $D$. Thus $\mathbb{E}_v \mathbb{E}_{Q_v}[|\widetilde{f}^{>\ell}|] \leq \gamma/\delta = \mathbb{E}_D[|\widetilde{f}^{>\ell}|]$. We will now show that $\chi^2(D, P) < \infty$, which would suffice for our result. Observe that

$$\chi^2(D, P) := \int_{y \sim R} \int_x \frac{R^2(y) D_y^2(x)}{R(y) G(x)} dx dy$$

$$= \int_{y \sim R} R(y) \int_x \frac{D_y^2(x)}{G(x) dx} = \int_{y \sim R} R(y) \chi^2(D_y, \mathcal{N}(0, \mathbf{I}_d)).$$

Observe that $D_y$ is obtained from $P_v^{\widetilde{A}_y}$ where $\widetilde{A}$ is supported only on $\{x : |x| \leq d\}$. [DKRS23, Lemma 3.1] implies that $\chi^2(D_v, \mathcal{N}(0, \mathbf{I}_d))$ is uniformly upper bounded by $O_d(1)$. Integrating this uniform upper bound by $O_d(1)$, we get the desired conclusion of $\chi^2(D, P) < \infty$.

## B.9 Formal version of Theorem 1.6

We are now ready to state and prove the formal version of Theorem 1.6.

**Theorem B.6** (SQ Hardness of Testing Problem 1.3). *Consider the testing problem in Testing Problem 1.3 with $E = \mathrm{DG}\big[0, \sigma, 0, s\big]$ for $s \asymp \alpha$ and $\sigma = 1$. Furthermore, assume that*

- $\alpha \gg \frac{1}{d^{\mathrm{polylog}(d)}}$ *(i.e., it is not too tiny)*

- $\rho^2 \asymp \alpha^2 \mathrm{polylog}(d/\alpha)$ *and $\rho \leq \rho_0$ for a sufficiently small absolute constant $\rho_0$.*

*Then we have the following guarantees:*

1. $\mathbb{P}_E(z = 0) \geq \alpha$ *(i.e., it is a valid instance).*

2. *Any SQ algorithm that solves the testing problem with probability at least $2/3$ either uses $q \gtrsim q_0 := d^{\log^2(d/\alpha)}$ many queries or uses a single query which is as powerful as VSTAT($m$) for $m \gtrsim \frac{\sqrt{d}}{\alpha^2 \mathrm{polylog}(d, 1/\alpha)}$.*

*Proof.* Since $\sigma \geq 1/2$, we get that $\mathbb{P}(z = 0) \geq \alpha$ (recall that $\mathbb{P}_{Z \sim \mathrm{DG}\big[0, \sigma, 0, s\big]}(z = 0) = \Theta(s/\sigma)$), which satisfies the first claim of Theorem B.6.

To establish the second claim about the SQ complexity, using Proposition 2.6, it suffices to show that the probability of success of $f$ on distinguishing $Q_v$ and $P$ with $m$ simulation complexity is at most $1/q_0$. Recall that the success event $\mathcal{E}_{f,v,m}$ is defined as the following event:

$$|\mathbb{E}_{Q_v}[f] - \mathbb{E}_P[f]| \geq \max\left(\frac{1}{m}, \min\left(\sqrt{\frac{(\mathbb{E}_P[f])(1 - \mathbb{E}_P[f])}{m}}, \sqrt{\frac{(\mathbb{E}_{Q_v}[f])(1 - \mathbb{E}_{Q_v}[f])}{m}}\right)\right), \quad (23)$$

and our goal is to show that for any fixed bounded query $f : \mathcal{Z} \to [0, 1]$, we have $\mathbb{P}_{v \sim \mathcal{S}^{d-1}}[\mathcal{E}_{f,v,m}] \leq \frac{1}{q}$ for $q \asymp d^{\log^2(d/\alpha)}$ and $m \gtrsim m_0 := \frac{\sigma^2 \sqrt{d}}{\rho^2 \mathrm{polylog}(d/\alpha)}$.

We now define the following events:

- First, $\mathcal{E}'_{f,v,m}$ is defined as: $\big|\mathbb{E}_{Q_v}[f(x, y)] - \mathbb{E}_{T_v}[f(x, y)]\big| \geq \frac{1}{4m^2}$.

- Next, the event $\mathcal{E}''_{f,v,m}$ is defined as: for a large constant $C$ (which can be deduced from the proof of Claim B.7),

$$\big|\mathbb{E}_{T_v}[f] - \mathbb{E}_P[f]\big| \geq \max\left(\frac{1}{Cm}, \min\left(\sqrt{\frac{(\mathbb{E}_P[f])(1 - \mathbb{E}_P[f])}{Cm}}, \sqrt{\frac{(\mathbb{E}_{T_v}[f])(1 - \mathbb{E}_{T_v}[f])}{Cm}}\right)\right).$$

Next, we show in Claim B.7 that $\mathcal{E}_{f,v,m} \subset \mathcal{E}'_{f,v,m} \cup \mathcal{E}''_{f,v,m}$. By the union bound and Claim B.7, it suffices to establish that the probabilities of these events individually is at most $\frac{1}{2q}$.

- ($\mathcal{E}'_{f,v,m}$) Theorem 3.7 implies the desired bound for any $m \leq (d/\alpha)^{\log^2(d/\alpha)}$ and $q \leq d^{\log^2(d/\alpha)}$.

- ($\mathcal{E}'_{f,v,m}$) This inequality was established in Proposition 3.6 for any $m \lesssim m_0$ with $m_0 \asymp \frac{\sigma^2 \sqrt{d}}{\rho^2 \sqrt{\log(1/q)}}$. Taking $q = d^{\log^2(d/\alpha)}$ and $\sigma = \Theta(1)$ leads to $m_0 \asymp \frac{\sqrt{d}}{\rho^2 \mathrm{polylog}(d/\alpha)}$.

This completes the proof of Theorem B.6. $\qquad\square$

We now provide the statement and the proof of Claim B.7.

**Claim B.7.** *We have that $\mathcal{E}_{f,v,m} \subset \mathcal{E}'_{f,v,m} \cup \mathcal{E}''_{f,v,m}$.*

*Proof of Claim B.7.* Indeed, we have that

$$\big|\mathbb{E}_{Q_v}[f(x,y)] - \mathbb{E}_P[f(x,y)]\big|$$

$$\leq \big|\mathbb{E}_{Q_v}[f(x,y)] - \mathbb{E}_{Q'_v}[f(x,y)]\big| + \big|\mathbb{E}_{Q'_v}[f(x,y)] - \mathbb{E}_P[f(x,y)]\big|$$

$$\leq \frac{1}{4m^2} + \max\left(\frac{1}{Cm}, \min\left(\sqrt{\frac{(\mathbb{E}_P[f])(1-\mathbb{E}_P[f])}{Cm}}, \sqrt{\frac{(\mathbb{E}_{T_v}[f])(1-\mathbb{E}_{T_v}[f])}{Cm}}\right)\right).$$

Observe that on $\mathcal{E}'_{f,v,m}$, $|\mathbb{E}_{T_v}[f] - \mathbb{E}_{T_v}[f]| \leq \tau$ for $\tau = O(1/m^2)$. Since the expectations are close, the standard deviations are also close: $|\sqrt{(\mathbb{E}_{T_v}[f])(1-\mathbb{E}_{T_v}[f])} - (\mathbb{E}_{Q_v}[f])(1-\mathbb{E}_{Q_v}[f])| = O(\sqrt{\tau})$. Therefore, the second term above is at most

$$\max\left(\frac{1}{Cm}, \min\left(\sqrt{\frac{(\mathbb{E}_P[f])(1-\mathbb{E}_P[f])}{Cm}}, \sqrt{\frac{(\mathbb{E}_{Q_v}[f])(1-\mathbb{E}_{Q_v}[f])}{Cm}}\right)\right) + \sqrt{O(\tau)}.$$

Since $\tau \gtrsim 1/m^2$, the overall term is at most

$$\frac{1}{Cm} + \max\left(\frac{1}{Cm}, \min\left(\sqrt{\frac{(\mathbb{E}_P[f])(1-\mathbb{E}_P[f])}{Cm}}, \sqrt{\frac{(\mathbb{E}_{Q_v}[f])(1-\mathbb{E}_{Q_v}[f])}{Cm}}\right)\right).$$

We now claim that this is less than the threshold for $\mathcal{E}_{f,v,m}$ in (23). Towards that goal, define $a = \sqrt{\mathbb{E}_P[f] \cdot \mathbb{E}_P[1-f]}$ and $b$ for the corresponding term with $Q_v$. Consider the case when $\min(a,b)/\sqrt{Cm} \leq 1/Cm$. Then the left hand side above is $\frac{2}{Cm}$, which is less than the quantity in $\mathcal{E}_{f,v,m}$, which is at least $1/m$. Suppose now that $\min(a,b) \geq 1/(Cm)$. Then the term above is at most $\frac{1}{Cm} + \frac{\min(a,b)}{\sqrt{Cm}} \leq 2\frac{\min(a,b)}{Cm}$, which is less than the quantity in $\mathcal{E}_{f,v,m}$, which is at least $\frac{\min(a,b)}{m}$.

Thus, we have shown that $\mathcal{E}_{f,v,m} \subset \mathcal{E}'_{f,v,m} \cup \mathcal{E}''_{f,v,m}$.

$\square$

## C  Computationally-Efficient Reduction from Testing to Estimation

Suppose there is an algorithm $\mathcal{A}$ with the following guarantees: given $n$ i.i.d. samples $(x_1, y_1), \ldots, (x_n, y_n)$ in $\mathbb{R}^d \times \mathbb{R}$ from Definition 1.1 with inlier probability $\alpha$ and regressor $\beta \in \mathbb{R}^d$, computes an estimate $\widehat{\beta}$ such that $\|\widehat{\beta} - \beta\|_2 \lesssim \tau$.

Consider the following (randomized) algorithm $\mathcal{A}'$ that takes $2n$ samples $S = \{(x_1, y_1), \ldots, (x_n, y_n)\}$ and $S' = \{(x'_1, y'_1), \ldots, (x'_n, y'_n)\}$ and perform the following operation:

- Sample a random rotation matrix $\mathbf{U} \in \mathbb{R}^{d \times d}$.

- Let $\widehat{\beta}_1$ be the output of $\mathcal{A}$ on $S$.

- Define $S'' := \{(\mathbf{U}x'_1, y'_1), \ldots, (\mathbf{U}x'_n, y'_n)\}$

- Let $w$ be the output of $\mathcal{A}$ on $S''$.

- Let $\widehat{\beta}_2 = \mathbf{U}^\top w$.

- Let $W = \left\langle \frac{\widehat{\beta}_1}{\|\widehat{\beta}_1\|}, \frac{\widehat{\beta}_2}{\|\widehat{\beta}_2\|_2} \right\rangle$. If $|W| > 1/9$, output "alternate", otherwise output "null".

**Theorem C.1.** *If $\tau \leq \rho/4$ and $d \gtrsim \log(1/\delta)$, then $\mathcal{A}'$ solves the testing problem in Testing Problem 1.3 with probability at least $1 - 2\delta$.*

*Proof.* We will argue the success probabilities separately.

**Alternate Distribution** Consider the case when the underlying distribution is alternate and let the latent hidden direction be $v$. Conditioned on $v$ and $\mathbf{U}$, the samples $S$ and $S''$ satisfy the conditions of Definition 1.1 with the underlying regressor $\beta$ and $\mathbf{U}\beta$, where $\beta := \rho v$; here we use that Gaussian distribution is rotationally invariant and $\mathbf{U}x$ is again distributed as isotropic Gaussian. Thus, the guarantees of $\mathcal{A}$ imply that with probability $1 - 2\delta$, we have that $\|\widehat{\beta}_1 - \beta\|_2 \leq \tau$ and $\|\widehat{\beta}_2 - \beta\|_2 = \|w - \mathbf{U}\beta\|_2 \leq \tau$. Since $\tau \leq \rho/4$, we have that $\widehat{\beta}_1\|_2 \leq 1.5\rho$ and the same for $\|\widehat{\beta}_2\|$. Since

$$\left\langle \widehat{\beta}_1, \widehat{\beta}_2 \right\rangle - \langle \beta, \beta \rangle = -\left\langle \widehat{\beta}_1 - \beta, \widehat{\beta}_2 - \beta \right\rangle - \langle \beta, \widehat{\beta}_2 - \beta \rangle - \langle \beta, \widehat{\beta}_1 - \beta \rangle,$$

the closeness guarantee implies that

$$\left| \left\langle \widehat{\beta}_1, \widehat{\beta}_2 \right\rangle - \langle \beta, \beta \rangle \right| \leq \tau^2 + 2\rho\tau \leq 3\rho^2/4.$$

Hence, with probability $1 - 2\delta$, we have that $|W| \geq (\rho^2/4)/(3\rho/2)^2 \geq 1/9$, and hence the algorithm would correctly output "alternate".

**Null Distribution** We will argue that $w$ is independent of $\mathbf{U}$. Indeed, for any $\mathbf{U}$, the distribution of the samples in $S''$ is i.i.d. from $\mathcal{N}(0, \mathbf{I}_d) \times R$, where $R$ is the marginal distribution of $y$ (recall that $y$ is independent of $X$ under the null). Hence, $S''$ and $w$ are independent of $\mathbf{U}$. Therefore, $\frac{\widehat{\beta}_2}{\|\widehat{\beta}_2\|_2}$ is distributed uniformly over the unit sphere (independent of $\beta_1$). Hence, $W$ is distributed as the product of two unit vectors, implying that with probability $1 - \delta$, $|W| \lesssim \sqrt{\frac{\log(1/\delta)}{d}}$, and hence the algorithm correctly outputs "null" for $d$ large enough. $\qquad \square$

# D    Inefficient SQ Algorithm with Correct Sample Complexity

In this section, we mention an SQ algorithm that uses $q = \exp(\widetilde{O}(d/\tau\alpha))$ queries from VSTAT$(m)$ with $m = \Theta(1/\alpha)$ and outputs an estimate $\widetilde{\beta}$ such that $\|\widehat{\beta} - \beta\|_2 \lesssim \tau$. Furthermore, this SQ algorithm can be simulated from $O\left(\frac{d \log(1/\alpha)}{\alpha}\right)$ i.i.d. samples from distribution $P_{\beta^*, E}$.

**Theorem D.1.** *Let $\|\beta^*\|_2 \leq 1$ and $\alpha \in (0, 1)$ and let the underlying distribution be $P_{\beta^*, E}$ for an unknown $E$ and known $\alpha$. There exists an SQ algorithm that uses $q \leq \exp(O(d \log(1/\tau\alpha)))$ many queries to VSTAT$(m)$ for $m \lesssim 1/\alpha$ and outputs an estimate $\widetilde{\beta}$ such that $\|\widehat{\beta} - \beta^*\| \lesssim \tau$.*

*Furthermore, with high probability, the VSTAT$(m)$ oracle for this SQ algorithm can be simulated using $m' = \widetilde{O}\left(\frac{d}{\alpha}\right)$ many i.i.d. samples from $P_{\beta^*, E}$.*

*Proof.* Let $\mathcal{C}$ be a $\tau'$-cover of $\{x : \|x\|_2 \leq 1\}$ with respect to the Euclidean norm for $\tau' = 0.01\tau\alpha$. We know such a cover exists with $\log |\mathcal{C}| \lesssim d \log(1/\tau')$. Furthermore, let $\beta' \in \mathcal{C}$ be $\tau'$-close to $\beta^*$. For each $\beta \in \mathcal{C}$, define the query $f_\beta(x, y) = \mathbb{1}_{|x^\top \beta - y| \leq \tau'}$.

The SQ algorithm is as follows:

> - For each $\beta \in \mathcal{C}$, let $v_\beta \leftarrow \text{VSTAT}(f_\beta, m)$.
>
> - Output $\widehat{\beta} = \text{argmax}_{\beta \in \mathcal{C}} v_\beta$.

**Correctness.** To show correctness, we shall show that for $\beta$ that is $\tau$-far from $\beta^*$, it must be the case that $v_\beta < v_{\beta'}$, which would imply that any such $\beta$ can not be the output.

Let us start by analyzing $\mathbb{E}[f_\beta]$. Let the distribution $E$ be $\alpha\delta_0 + (1 - \alpha)E'$ for an arbitrary distribution $E'$, where $\delta_0$ is the point mass at origin. Then observe that for $G \sim \mathcal{N}(0, 1)$:

$$\mathbb{E}[f_\beta] = \alpha\mathbb{E}_x[\mathbb{1}_{|x^\top(\beta - \beta^*)| \leq \tau'}] + (1 - \alpha)\mathbb{E}_{x, z \sim E'}[\mathbb{1}_{|x^\top(\beta - \beta^*) + z| \leq \tau'}]$$
$$= \alpha\mathbb{P}(|G| \leq \tau'/\|\beta - \beta^*\|) + (1 - \alpha)\mathbb{P}_{G, z \sim E'}(|G \cdot \|\beta - \beta^*\|_2 + z| \leq \tau').$$

In particular, for $\beta'$, $\mathbb{E}[f_{\beta'}] \geq 0.5\alpha$ because $\|\beta' - \beta^*\| \leq \tau'$ and $\mathbb{P}(|G| \leq 1) \geq 0.5$. It can then be checked $\max_{\beta \in \mathcal{C}} \geq v_{\beta'} \geq \mathbb{E}[f_{\beta'}] - \frac{1}{m} - \sqrt{\frac{\mathbb{E}[f_{\beta'}]}{m}}$, which is bigger than $\mathbb{E}[f_{\beta'}]/2$ if $m \gtrsim \frac{1}{\mathbb{E}[f_{\beta'}]}$, which is satisfied since $m \geq \frac{1}{\alpha}$ and $\mathbb{E}[q_\beta] \geq 0.5\alpha$.

Now consider any $\beta$ such that $\|\beta - \beta^*\| = r \geq \tau = 100\tau'/\alpha$. Then

$$\mathbb{E}[f_\beta] = \alpha \mathbb{P}(|G| \leq \tau'/r) + (1-\alpha) \mathbb{P}_{G, z \sim E'}(|Gr + z| \leq \tau')$$
$$\leq \alpha \mathbb{P}(|G| \leq \tau'/r) + (1-\alpha) \max_{z' \in \mathbb{R}} \mathbb{P}_G(|Gr + z'| \leq \tau')$$
$$\leq \frac{\alpha\tau'}{r} + (1-\alpha)\frac{\tau'}{r}$$
$$\leq \frac{\tau'}{r} \leq 0.01\alpha.$$

Therefore, for any such $\beta$, $v_\beta \leq \mathbb{E}[f_{\beta'}] + \frac{1}{m} + \sqrt{\frac{\mathbb{E}[f_{\beta'}]}{m}} \leq 0.02\alpha$ if $m \gtrsim 1/\alpha$. Therefore, any such $\beta$ can not be $\widehat{\beta}$ and hence $\|\widehat{\beta} - \beta^*\|_2 \leq \tau$.

**Simulation with samples** We implement the VSTAT($m$) oracle by taking a set $S$ of i.i.d. samples and returning the empirical mean of $q_\beta$ over $S$. Observe that all of the queries $f_\beta$ are halfspaces and hence have VC Dimension $O(d)$. For $i \in \{1, \ldots, \log(1/\alpha_0)\}$, let $\mathcal{A}_i = \{\beta : \mathbb{E}[q_\beta] \in [2^i\alpha, 2^{i+1}\alpha] \cup [1 - 2^{i+1}\alpha, 1 - 2^i\alpha]\}$. Let $\mathcal{A}_0 = \{\beta : \mathbb{E}[q_\beta] \in [0, \alpha] \cup [1 - \alpha, 1]\}$.

By uniform concentration [BLM13, Theorem 13.7] and [BLM13, Theorem 12.5], if $n \geq \frac{d \log(1/2^{i+1}\alpha)}{2^{i+1}\alpha}$, then with probability $1 - \delta/J$ for $J = \log(1/\alpha)$, for all $\beta \in \mathcal{A}_i$ for $i \in \mathbb{N} \cup \{0\}$, we have

$$\left|\mathbb{E}_S[f_\beta] - \mathbb{E}_{P_{\beta^*,E}}[q_\beta]\right|$$
$$\lesssim \sqrt{2^i\alpha}\sqrt{\frac{d \log(1/2^i\alpha)}{n}} + \sqrt{2^i\alpha} \cdot \sqrt{\frac{\log(J/\delta)}{n}} + \frac{\log(J/\delta)}{n}$$
$$\lesssim \sqrt{\mathbb{E}_{P_{\beta^*,E}}[q_\beta] \cdot (1 - \mathbb{E}_{P_{\beta^*,E}}[q_\beta])} \cdot \sqrt{\frac{dJ + \log(J/\delta)}{n}} + \sqrt{\alpha} \cdot \sqrt{\frac{dJ + \log(J/\delta)}{n}} + \frac{\log(J/\delta)}{n},$$

where we use that $\alpha + \mathbb{E}[f_\beta] \cdot (1 - \mathbb{E}[f_\beta]) \gtrsim 2^i\alpha$ for all $i \in \mathbb{N} \cup \{0\}$.

By a union bound over $\mathcal{A}_i$'s, this uniform concentration holds for all $\beta \in \mathbb{R}^d$. That is, if $n \geq \frac{dJ + \log(J/\delta)}{\alpha}$, then with probability $1 - \delta$, for all $\beta \in \mathbb{R}^d$, we have

$$\left|\mathbb{E}_S[f_\beta] - \mathbb{E}_{P_{\beta^*,E}}[q_\beta]\right|$$
$$\lesssim \sqrt{\mathbb{E}_{P_{\beta^*,E}}[q_\beta] \cdot (1 - \mathbb{E}_{P_{\beta^*,E}}[q_\beta])} \cdot \sqrt{\frac{dJ + \log(J/\delta)}{n}} + \sqrt{\alpha} \cdot \sqrt{\frac{dJ + \log(J/\delta)}{n}} + \frac{\log(J/\delta)}{n}$$
$$\leq \sqrt{\mathbb{E}_{P_{\beta^*,E}}[q_\beta] \cdot (1 - \mathbb{E}_{P_{\beta^*,E}}[q_\beta])} \cdot \sqrt{\frac{1}{m}} + \frac{1}{m},$$

if $n \gtrsim (dJ + \log(J/\delta)) \cdot (m + \alpha m^2) + m \log(J/\delta)$. On this event, we get that the empirical approximation is a VSTAT($m/4$) oracle. Since we need $m = \Theta(1/\alpha)$, the required sample complexity for failure probability $\delta$ is at most $\frac{d \log(1/\alpha) + \log(\log(1/\alpha)/\delta)}{\alpha}$.

$\square$

# E    Efficient SQ Algorithm with Matching Accuracy

We now show that there exists an efficient SQ algorithm that solves Definition 1.1 and the hard instance in Theorem 1.6 with polynomially number of VSTAT($d/\alpha^2$) queries. Let $\beta^*$ be the unknown regressor with $\|\beta^*\|_2 \leq 1$. In this section, we use $u$ as a shorthand for $(x, y)$.

**Theorem E.1.** *Let $\alpha \in (0, 1)$ and $\beta^* \in \mathcal{B}$, where $\mathcal{B} := \{\beta : \|\beta\|_2 \leq 1\}$. For any $\epsilon \in (0, 1)$, there is an SQ algorithm that takes these $\alpha, \epsilon$ as input, makes $\mathrm{poly}(d)$ number of queries to VSTAT $\left(\frac{d}{\epsilon\alpha^2}\right)$ on $P_{\beta^*,E}$, and (iii) computes an estimate $\widehat{\beta} \in \mathbb{R}^d$ such that $\|\widehat{\beta} - \beta^*\|_2 \lesssim \epsilon\alpha$.*

Observe that we do not need $\epsilon$ to be very small to solve the hard instance of Testing Problem 1.3, i.e., we can set $\epsilon = \rho/\alpha = \widetilde{\Theta}(1)$ and still solve Testing Problem 1.3 with polynomial number of queries to VSTAT$(\widetilde{\Theta}(d/\alpha^2))$.

*Proof.* Define the function $g(x) : \mathcal{X} \to \{0, 1\}$ to be function such that $g(x) = 0$ if and only if $\|x\|_2 \geq L\sqrt{d}$ for $L = \mathrm{polylog}(d/\alpha\epsilon)$. Consider the loss function $\ell(\beta, u) := g(x) \cdot \ell_{\mathrm{Huber}}\left(y - x^\top \beta\right)$; here $\ell_{\mathrm{Huber}}(\cdot)$ is the Huber loss with the gradient $h(z) = z\mathbb{1}_{z \in [-1,1]} + \mathrm{sgn}(z)\mathbb{1}_{|z|>1}$. Consider the averaged loss $\mathcal{L}(\beta) := \mathbb{E}_{u \sim P_{\beta^*, E}}[\ell(\beta, u)]$.

We claim the following:

1. $\mathcal{L}$ is $\kappa$-strongly convex on $\mathcal{B}$ for $\kappa = \Theta(\alpha)$.

2. $\mathcal{L}$ is $L_1$ smooth (Lipschitz continuous gradient) on $\mathcal{B}$ for $L_1 = O(1)$.

3. For every $z \in \mathcal{Z}$, the function $\ell(\cdot, z)$ is convex, and it is $L_0$-Lipschitz for $L_0 \lesssim L\sqrt{d}$.

4. $\beta^*$ is the unique minimizer of $\mathcal{L}$.

Therefore, we can apply [FGV17, Corollary 4.12] with parameters $L_0$, $L_1$, and $\kappa$ to find an $\alpha\epsilon$-close estimate $\widehat{\beta}$ such that $\|\widehat{\beta} - \arg\min \mathcal{L}(\beta)\|_2 \lesssim \alpha\epsilon$ with $O\left(\frac{dL_1 \log(L_1 \mathrm{diam}(\mathcal{B})/\alpha\epsilon)}{\kappa}\right) = O\left(\frac{d \log(1/\alpha\epsilon)}{\alpha}\right)$ many queries to VSTAT$\left(O\left(\frac{L_0^2}{\alpha\epsilon\kappa}\right)\right) = $ VSTAT$\left(\frac{d \cdot \mathrm{polylog}(d/\alpha)}{\alpha^2\epsilon}\right)$. We get the desired conclusion by noting that $\beta^*$ uniquely minimizes $\mathcal{L}(\beta)$.

We now give the details omitted earlier:

1. For any unit vector $v$, $v^\top \nabla^2 \mathcal{L} v$ is equal to $\mathbb{E}_u[g(x)\nabla^2 \ell_{\mathrm{Huber}}(y - x^\top \beta)(x^\top u)^2]$. The convexity follows by non-negativity of the Huber loss.

$$v^\top \nabla^2 \mathcal{L} v = \mathbb{E}_u[g(x)\nabla^2 \ell_{\mathrm{Huber}}(y - x^\top \beta)(x^\top v)^2] = \mathbb{E}_u[g(u)\mathbb{1}_{|y-x^\top\beta|\leq 1}(x^\top v)^2]$$
$$\geq \alpha \cdot \mathbb{E}_{x \sim \mathcal{N}(0, \mathbf{I}_d)}[g(x)\mathbb{1}_{|x^\top\beta^* - x^\top\beta|\leq 1}(x^\top v)^2] \gtrsim \alpha.$$

   The last inequality follows because $g(x)\mathbb{1}_{|x^\top\beta^* - x^\top\beta|\leq 1}(x^\top v)^2 \gtrsim \mathbb{1}_{\|x\|_2 \leq L\sqrt{d}}\mathbb{1}_{|x^\top w|\leq 1}\mathbb{1}_{|x^\top v|\geq 0.5}$ for some unit vector $w$. Using triangle inequality, we obtain that its probability is lower bounded by $\mathbb{P}[\mathbb{1}_{|x^\top w|\leq 1}\mathbb{1}_{|x^\top v|\geq 0.5}] - \mathbb{P}((1 - g(x))) \gtrsim 1 - d^{-100} \gtrsim 1$.

2. The smoothness follows from the same arguments as above by upper bounding $g(x)$ and $\nabla^2 \ell_{\mathrm{Huber}}$ by 1.

3. Observe that the gradient satisfies $\nabla\ell(\beta, z) = g(x)h(y - x^\top \beta)x$ and therefore $\|\nabla\ell(\beta, z)x\|_2 \leq L\sqrt{d}$, where we use that $\|xg(x)\| \leq \sqrt{L}d$ and the gradient of Huber loss is bounded by 1.

4. By strong convexity on $\mathcal{B}$, it suffices to show that $\beta^*$ has zero gradient.

$$\|\nabla\mathcal{L}(\beta^*)\|_2 = \|\nabla\mathbb{E}[g(x)h(z)x]\|_2 = 0,$$

   where we use that $xg(x)$ is a symmetric random variable and independent of $z$.

$\square$

