# OpenReview forum: "Information-Computation Tradeoffs  for Noiseless Linear Regression with Oblivious Contamination"
_NeurIPS.cc/2025/Conference — NeurIPS 2025 poster_

### Official Review · Reviewer_xzbN · 2025-06-30

**Clarity:** 3
**Significance:** 3
**Originality:** 3
**Rating:** 5
**Confidence:** 4

**Summary:**

This paper studies the noiseless linear regression with oblivious contamination in the response variable. Under isotropic gaussian design, if (1-alpha) fraction of the responses is obviously corrupted, it is well known that d/alpha sample is needed to recover the regression coefficient. However all the poly time algorithms require d/alpha^2 samples. This paper proves lowerbound showing that any SQ algorithm either requires polynomial sqrt(d)/alpha^2 sample or superpolynomial time. Previous work [DKS17;DKS19] on the SQ lower bound for linear regression does not apply to the setting because the “noise” distribution has to include an alpha point mass at 0. Therefore following the work of DKRS23, a discrete gaussian distribution is used as the noise. Then the testing hardness is decomposed into two parts: 1) testing hardness between conditional gaussian 2)testing hardness between gaussian and discrete gaussian. The first part is straightforward and has been similarly proven in the prior work. Leveraging the fact that the low order Hermite moment of discrete gaussian and gaussian distribution nearly matches, they proved the second part and completed the proof.

**Questions:**

No

**Ethical Concerns:**

["NO or VERY MINOR ethics concerns only"]

**Final Justification:**

There was no discussion during rebuttal and I will maintain my score of Accept.

**Limitations:**

Yes

**Quality:**

3

**Strengths And Weaknesses:**

Strength:
The paper addresses a well known open problem and suggests that the sample complexity difference between efficient and inefficient algorithm in the noiseless linear regression with oblivious contamination is fundamental.

Weakness:
The paper is mostly well written and easy to follow. However, I find page 8 and page 9 hard to read and the proof sketches hard to follow especially when some external lemmas are cited. I would suggest rewriting the proof sketch and potentially removing some of them.
Most of the proof techniques are borrowed from DKRS23.

---

> ### Author Rebuttal · Authors · 2025-07-31
>
> We thank the reviewer for their time and effort in providing feedback and their positive assessment of our work.
>
> Regarding the proof sketches on Pages 8 and 9: Unfortunately the page restriction did not give us sufficient room for proper explanations of everything in the main body; we will reorganize/rephrase them better using the additional page allowed in the revised version.

---

> > ### Comment · Reviewer_xzbN · 2025-08-05
> >
> > Thanks for the response!

---

### Official Review · Reviewer_jfnB · 2025-07-03

**Clarity:** 3
**Significance:** 3
**Originality:** 3
**Rating:** 4
**Confidence:** 2

**Summary:**

This paper studies the linear regression problem, where the noise is i.i.d. but sparse since it has a mass at $0$ controlled by $\alpha$. The paper shows that the for SQ algorithms, the number of samples required is proportional to $\frac{1}{\alpha^2}$, which matches the scaling in $\alpha$ for IT bounds.

**Questions:**

1. Why did we choose to analyze SQ algorithms instead of another class of algorithms? Is it the most likely candidate apriori to veryfiy Question 1.2?

2. How sensitive are the hardness results to the precise distribution of the contamination? For example, if the contamination was not discrete Gaussian how would the number of samples change?

3. Is there intution of why a discrete Gaussian was chosen instead of another distribution? Is the choice of the discrete Gaussian optimal in the sense that it gives the largest sample complexity?

**Ethical Concerns:**

["NO or VERY MINOR ethics concerns only"]

**Final Justification:**

I agree with the other reviewers. This is nice work and I maintain my original positive assessment of this paper.

**Limitations:**

Yes

**Paper Formatting Concerns:**

No issues

**Quality:**

3

**Strengths And Weaknesses:**

The paper is well written and provides a nice review of relevant notions for a general audience. The problem is clear and the steps in the outline of the proof are well explained, so the main result is belieavable. This is done by reducing the problem to a form that allows applications of results from a series of papers by Diakonikolas et al.

The question of the sharpness of the sample complexity is very natural, and the main result provides convincing evidence on that the sample complexity bounds are indeed sharp for a class of natural algorithms. There seems to be significant differences in this problem so that it does not follow as a simple corollary of known results, and new ideas in the gradual reduction of the problem is needed.

---

> ### Author Rebuttal · Authors · 2025-07-31
>
> We thank the reviewer for their time and effort in providing feedback and their positive assessment of our work.
>
> - We analyzed the class of SQ algorithms because it is a very versatile and powerful class of algorithms. Please see \[FGRVX17] for details. We believe that the same construction should lead to similar hardness against other classes of algorithms, such as low-degree polynomial tests (see \[BBHLS21] for a connection to SQ algorithms) and natural Sum-of-Square formulations.
>
> - Sensitivity to the precise distribution of the contamination:
>
>   - The sample complexity requirement is understandably dependent on the choice of the noise distribution. Intuitively, if the noise is very benign (for example, if the noise is always zero), then roughly $d$ samples suffice (both information-theoretically and computationally), whereas if the noise is chosen carefully (say discrete Gaussian, as we show), then the computational sample complexity can be large. Importantly, we prove an information-computation tradeoff over the worst-case oblivious contamination, which is the standard goal in the minimax setting.
>
>   - The SQ hardness would continue to hold not only for discrete Gaussians (DG), but any distribution $E$ that is sufficiently close to DG in terms of its moments. This follows from the arguments implicit in Theorem 3.7, where we transferred SQ hardness from usual Gaussian to DG using nearly matching moments.
>
> - The choice of discrete Gaussian
>   - At a high level, we chose discrete Gaussian because of the following insights (please see Section 1.2 for more details)
>     - The case of continuous Gaussian is information-theoretically and SQ hard (but, unfortunately, is not a valid instance of our problem as it does not assign $>= \alpha$ mass to 0), and
>     - Distinguishing between continuous Gaussian and discrete Gaussian should be SQ-hard.
> Hence, by choosing discrete Gaussian as our instance, we simultaneously satisfy the model assumption and the desired SQ
>   - Furthermore, the discrete gaussian noise is also optimal (in the sense of the largest SQ complexity) for the \*testing\* problem: there exists an efficient SQ algorithm that solves the testing problem (with any valid noise satisfying Definition 1.1) with only polynomially many queries to the VSTAT$(m)$ oracle for $m$ roughly $\sqrt{d}/\alpha^2$.  As mentioned in the conclusions section (Section 4), establishing optimality for the estimation problem is left as an interesting open problem.

---

> > ### Comment · Reviewer_jfnB · 2025-08-05
> >
> > Thank you for the clarifications. I am less familiar with this line of work, but I agree with the other reviewers. This is nice work.

---

### Official Review · Reviewer_1Aum · 2025-07-05

**Clarity:** 2
**Significance:** 3
**Originality:** 3
**Rating:** 5
**Confidence:** 4

**Summary:**

This paper studies robust linear regression in the oblivious noisy model.
Namely, we observe $y, X$ where the rows of $X$ are iid standard normal (in $d$ dimensions), and $y=X\beta+z$ is an $n$ vector. Further the entries of $z$ are i.i.d independent of $X$, with a fraction $\alpha$ equal to $0$. We would like to reconstruct exactly the $d$-dimensional vector $\beta$.
Information theoretically, exact reconstruction can be achieved  as soon as $n\ge C d/\alpha$. On the other hand, all known polytime algorithms require at least $C'd/\alpha^2$ entries. This paper provides rigorous evidence towards an information computation gap in this model. It proves that, wthin the statistical query model, exact recovery requires either a superpolynomial number of queries or sample size at least d^{1/2}/\alpha^2.

**Questions:**

The proof goes through multiple equivalent rephrasings of the same testing problems. Can this be simplified?

**Ethical Concerns:**

["NO or VERY MINOR ethics concerns only"]

**Limitations:**

Yes.

**Quality:**

3

**Strengths And Weaknesses:**

Strengths: The problem is interesting and the result is the first one providing evidence of a gap. The proof is non-trivial.

Weaknesses: The result is not optimal and implies a gap only when $\alpha \ll 1/\sqrt{d}$.
The statistical query model has some limitations and the connection to the original problem is not as direct as one would like it to be.
The presentation of technical arguments is somewhat clumsy (not the argument themselves). For instance lines 81-83: Why do you write
"$\Theta$ is set to $P$ wher under $P$...".  Why not just "Under $\Theta$:..". Or $\Theta=P$ where $P$ is the distribution of:..."
or $..P=N(0,I_d)\otimes R^*_{\rho, E}...$.  The next two lines are even worse.

---

> ### Author Rebuttal · Authors · 2025-07-31
>
> We thank the reviewer for their time and effort in providing feedback and their positive assessment of our work. We respond to their questions in more detail below:
>
> - _“Connection to the original problem is not as direct as one would like to be”_
>
>   - We point out that our SQ-hardness result also applies directly to the search version of the problem (see \[FGRVX17, Definition 2.1] for details) without going to the testing problem. As an example, see \[DKPPS21] who studied both the search and testing versions.
>
>   - We presented the testing problem because (i) the testing problem already captures the underlying SQ hardness while being no harder than estimation, and (ii) the connection between SQ and low-degree polynomials has been established for testing problems \[BBHLS21].
>
>   - Please let us know if we have misunderstood your question.
>
> - _“It proves that, within the statistical query model, exact recovery requires …”_
>
>   - We would like to mention that we establish hardness not only for exact recovery, but also for weak recovery (error less than $\rho/4$). This follows from Proposition 1.4 and Theorem 1.6.
>
> - _“The proof goes through multiple equivalent rephrasings of the same testing problems. Can this be simplified?_
>
>   - We use only one rephrasing (Proposition 3.2), which follows from a simple computation of conditional distribution. Please let us know if we may have misunderstood your point.
>
>   - If the reviewer is asking whether we could directly prove SQ hardness for discrete Gaussians without the connection with standard Gaussian, then for the reasons mentioned in Section 1.2 (related to Gaussian Fourier Analysis), it is unclear to us.
>
> - _“The result is not optimal and implies a gap only when $\alpha \ll 1/\sqrt{d}$”_
>
>   - We agree with the reviewer, and we have mentioned this as a concrete open problem in the Conclusion section.

---

### Official Review · Reviewer_wMTA · 2025-07-17

**Clarity:** 3
**Significance:** 2
**Originality:** 2
**Rating:** 4
**Confidence:** 4

**Summary:**

The paper studies linear regression for Gaussian i.i.d features with unknown noise distribution. The problem exhibits a gap between the number of samples that are needed in theory and the number of samples that are needed by typical algorithms to work. In the fully Gaussian setting (Gaussian covariates with Gaussian noise), the problem is well understood. The paper derives a lower bound on the number of samples required by statistical query algorithms

**Questions:**

Section 1:

- line 29 “and is limited in their capability 29 by requiring the contamination be independent of the samples” —> perhaps “is limited by the fact that” or “with the limitation that”
- Definition 1.5. comes a little out of the blue and lines 98 - 100 are not very illuminating regarding definition. It would be much more interesting to know why the value v is important. I.e how can this value be used with respect to Testing Problem 1.3.
- lines 111-112, you use “or” twice
- line 111, you must add a line of explanation on the fact that the number of queries is superpolynomial in d. E.g. If you take log(d/a) = log_d(d/a)/log_d(e) you get ((d/a)^2)^(1/log_d(e)) = ((d/a)^2)^(log_e(d))

Section 1.2.

- lines 128 - 129, the sentence does not make any sense to me. I would rephrase like this: “In order to prove Theorem 1.6., we need to exhibit a distribution E that simultaneously satisfies (i) P(Z=0)\geq \alpha  and (ii) such that the AQ algorithm either uses d^… many samples or uses at least one query …”
- line 149 “x and y are independent with correct marginals” —> What do you mean?
- lines 151-152, Why is the distribution of (x, y) conditioned on y orthogonal to v?
- lines 150-151, What do you mean “Q_v is a standard Gaussian in the direction orthogonal to v and is given by some known distribution A_y in the v-direction”? This is not clear. If Q_v is a standard Gaussian, why do you then say it is given by a distribution A_y?
- lines 155 - 162 should go. They do not add anything to the clarity of the paper. Adding too much information without providing associated clear explanations does not make the paper clearer
- Same comment for lines 163 - 166.
- line 174, what is a large “SQ dimension”?
- From what I understand, the key difficulties are (i)
- liens 126 - 127, you start by mentioning distributions A_y and B_y and motivate the section as the connection between the discrete and continuous Gaussian distributions but then jumps back to Q_v and T_v without any explanation
- lines 184, 186 you use the expression “inverse super polynomially small” why not just give the order of the bound. This would really be more clear and you would spare space.
- The connection between the statement of Theorem 1.6. and the conclusion of section 1.2. I.e. the bound on |E_{Q_v} [f] - E_{P}[f]| is not clear. Definition 2.5 should come way earlier
- Generally speaking, I feel that in section 1.2.

Section 2

- lines 198- 199: can’t you make the line shorted “we use h_k to denote ….” —> “we let h_k = 1/ \sqrt{k} (-1)^k” —> the best way to communicate is the mathematical definition. The rest is unnecessary/redundant
- lines 267 - 271, the definition of the Discrete Gaussian should come earlier. It should appear when you start using the term (It is actually central to the paper as indicated by the proof of Theorem 1.6). Perhaps also the definition of the NGCA.

**Ethical Concerns:**

["NO or VERY MINOR ethics concerns only"]

**Limitations:**

see above

**Quality:**

3

**Strengths And Weaknesses:**

Although the paper is well written I also believe that it is not yet ready for publication. You spend a lot of the time on the research you did while all that matters is the final result (see line 155 - 162 for example. All of this should go. Focus on the main result of the paper and explain this result as clearly as possible.). I think you need some more time to write the paper more clearly


Here the crux of the paper is the construction of the distribution used in the proof of Theorem 1.6. How you build this distribution is by (1) investigating the Gaussian distribution because for this distribution you can show hardness of the detection problem then turn to the discrete Gaussian because your assumptions require the probability of 0 to be larger than alpha. From what I understand you then start from the discrete GD and try to show hardness of the detection problem for the discrete GD by relying on the mathematical tools (in particular Gaussian Fourier Analysis) developed for Non Gaussian Component Analysis and show that it is hard to distinguish between continuous and discrete Gaussian noise. Now those tools have to be adapted to your setting. If I don’t make too many mistakes, I think it would be good to have a summary of that form appear somewhere in the paper. Get rid of most of section 1.2. and replace this with a general clear sketch of the proof of Theorem 1.6. (remove all the difficulties you faced)

---

> ### Author Rebuttal · Authors · 2025-07-31
>
> We thank the reviewer for their time and effort in providing feedback and their positive assessment of our work.
>
> We are grateful to the reviewer for their suggestions and typos.
>
> Please find the answers to questions below:
>
> - Line 149: We mean that under $P$, the marginal of $X$ (respectively $y$) matches that of $X$ under $Q_v$ (respectively $y$ under $Q_v$) . But unlike $Q_v$, where $X$ and $y$ are not independent, they are independent under $P$.
>
> - Lines 151-152: We meant that the distribution of $X|y$ has the following structure (please see the proof of Proposition 3.3 for more details):
>
>   - If $u$ is orthogonal to $v$, then $u^\top X$ (conditioned on $y$) is distributed as standard Gaussian.
>
>   - If $u$ is equal to $v$, then $u^\top X$ (conditioned on $y$) is distributed as $A_y$.
>
> That is, $X|y$ is an instance of High-DImensional Hidden Direction Distribution (Definition 2.7). In the notation of that definition, $X|y \~ P_v^{A_y}$.
>
> - Line 174: We meant Definition 2.4. We will update the terminology and link the definition there.

---

### Decision · Program_Chairs · 2025-09-17

**Decision:**

Accept (poster)

**Comment:**

This paper establishes a new sample complexity lower bound for the problem of linear regression with outliers. While the lower bound does not exactly match the known upper bound, it helps understand the separation between noise-free and noisy regression problems. Reviewers found the paper sound and well-written, yet also raised concerns that some of key techniques have already been established in prior works. Overall, the strengths outweigh the weaknesses.